# Complementary Ribo-seq approaches map the translatome and provide a small protein census in the foodborne pathogen *Campylobacter jejuni*

Kathrin Froschauer [1,9], Sarah L. Svensson [1,8,9], Rick Gelhausen[2], Elisabetta Fiore[1], Philipp Kible [1], Alicia Klaude[3,4], Martin Kucklick [3,4], Stephan Fuchs [5], Florian Eggenhofer [2], Chao Yang [6], Daniel Falush[6], Susanne Engelmann[3,4], Rolf Backofen [2,7] & Cynthia M. Sharma [1] ✉

In contrast to transcriptome maps, bacterial small protein (≤50-100 aa) coding landscapes, including overlapping genes, are poorly characterized. However, an emerging number of small proteins have crucial roles in bacterial physiology and virulence. Here, we present a Ribo-seq-based high-resolution translatome map for the major foodborne pathogen *Campylobacter jejuni*. Besides conventional Ribo-seq, we employed translation initiation site (TIS) profiling to map start codons and also developed a translation termination site (TTS) profiling approach, which revealed stop codons not apparent from the reference genome in virulence loci. Our integrated approach combined with independent validation expanded the small proteome by two-fold, including CioY, a new 34 aa component of the CioAB oxidase. Overall, our study generates a high-resolution annotation of the *C. jejuni* coding landscape, provided in an interactive browser, and showcases a strategy for applying integrated Ribo-seq to other species to enrich our understanding of small proteomes.

A complete census of coding and non-coding genomic features is key to understanding how bacteria survive and adapt to environmental challenges. For pathogens, this can reveal genes and mechanisms by which they adapt to changing environments, cause disease, or resist antibiotics. RNA-sequencing (RNA-seq) has rapidly expanded our catalog of factors that might influence stress responses and infection in many bacteria, including hundreds of potential small non-coding regulatory RNAs (sRNAs)[1]. However, RNA-seq does not directly reveal translated regions. Small open-reading frames (sORFs) have been identified in unexpected places, such as in untranslated regions (UTRs) of mRNAs, on dual-function sRNAs, or even overlapping longer ORFs[2–4].

Many of these sORFs encode so-called 'small proteins', here defined as ribosomally-synthesized polypeptides ≤70 amino acids (aa).

[1]University of Würzburg, Institute of Molecular Infection Biology, Department of Molecular Infection Biology II, Würzburg, Germany. [2]Bioinformatics Group, Department of Computer Science, University of Freiburg, Freiburg, Germany. [3]Technische Universität Braunschweig, Institute for Microbiology, Braunschweig, Germany. [4]Helmholtz Centre for Infection Research (HZI), Braunschweig, Germany. [5]Robert Koch Institute, Methodenentwicklung und Forschungsinfrastruktur (MF), Berlin, Germany. [6]The Center for Microbes, Development and Health, CAS Key Laboratory of Molecular Virology and Immunology, Shanghai Institute of Immunity and Infection, Chinese Academy of Sciences, Shanghai, China. [7]Signalling Research Centre CIBSS, University of Freiburg, Freiburg, Germany. [8]Present address: The Center for Microbes, Development and Health, CAS Key Laboratory of Molecular Virology and Immunology, Shanghai Institute of Immunity and Infection, Chinese Academy of Sciences, Shanghai, China. [9]These authors contributed equally: Kathrin Froschauer, Sarah L. Svensson. ✉e-mail: cynthia.sharma@uni-wuerzburg.de

Due to their small size, they have long been overlooked or have even been discarded from bacterial genome annotations. However, the functional characterization of several examples suggests that they have key roles in diverse processes in bacteria, including virulence, and often bind and modulate the activity of regulators, enzymes, and transport complexes (for an overview of small protein-related phenotypes and mechanisms, see refs. 3,5,6).

Bacteria, thus, likely still hide additional small genes with diverse roles in physiology and virulence. While comparative genomics has revealed thousands of small protein candidates[7–9], their validation has been challenging. They are difficult to detect by mass spectrometry (MS), even with new methodologies tailored for short sequences (reviewed in ref. 10). To bridge this gap, the ribosome profiling (Ribo-seq) technique has been harnessed to map and measure the 'translatome' with high sensitivity as a proxy for protein expression[11]. In Ribo-seq, ~ 30 nucleotide(nt)-long mRNA 'footprints' that are protected by translating ribosomes from nuclease digestion are analyzed by deep sequencing, which identifies and precisely maps ORFs and measures translation genome-wide. Ribo-seq has revealed widespread translation of hidden, novel sORFs in eukaryotes and their viruses[12], archaea[13–15], as well as in Gram-negative and -positive bacteria[16–18] (reviewed in ref. 19).

Ribo-seq variations that enrich footprints at start codons by treatment with inhibitors of initiating ribosomes [e.g., the pleuromutilin retapamulin (Ribo-RET[20,21]) or the proline-rich antimicrobial peptide (PrAMP) oncocin 112[22]] have now been developed for some bacteria. These allow precise annotation of start codons (translation initiation sites/TIS) based on experimental data and facilitate the discovery of (s)ORFs within bacterial genes, such as alternative N-terminal isoforms or those nested out-of-frame within longer ORFs[4,19]. Alternative (s)ORFs can also arise from premature in-frame stop codons generated by mutation (e.g., phase variation) or frameshifting. Prediction of frameshift sites from genome sequences is not trivial, and experimental detection of translation termination sites (TTS) might reveal more examples. An *E. coli* Ribo-seq dataset that enriched ribosomes at stop codons with the PrAMP apidaecin (Api), which targets terminating ribosomes, has been used to increase the confidence in novel sORFs[23]. However, TTS has not yet been used to comprehensively refine translatome maps, and outside of *E. coli*, such datasets are not yet available. In contrast to refined transcriptome annotations (transcription start sites, processing, novel sRNAs) based on diverse RNA-seq methods, which have been invaluable for studying the regulation and functional genomics of diverse species, integrated translatomics datasets that provide comprehensive information for ORFs of all sizes are lacking for most bacterial species, including many pathogens. This includes *Campylobacter jejuni*, currently the most common cause of bacterial gastroenteritis.

Our primary transcriptome maps for four strains of this widespread foodborne pathogen revealed strain-specific promoter usage, potential variation in ORF lengths, as well as conserved and infection-relevant sRNAs[24–27]. In contrast, the *C. jejuni* translatome and small ORFome have not yet been examined. Knowing the whole gene repertoire for *C. jejuni* will provide a basis to better understand its physiology and virulence, as still very little is known about how it causes diseases and regulates its gene expression[28]. Here, we provide a comprehensive translatome map and re-annotation of the *C. jejuni* protein-coding capacity based on three complementary Ribo-seq methods, including a small protein catalog. In addition to 'standard' Ribo-seq and TIS profiling, we also employed TTS profiling using Api and sequencing of disome footprints. We leveraged TTS data to reveal stop codon usage not apparent from the reference genome (e.g., in phase-variable ORFs). We independently validate the translation of 47 out of 55 annotated sORFs ≤ 70 aa using western blotting and MS and use this to guide the Ribo-seq-based discovery of 42 new high-confidence sORFs in diverse genomic contexts, of which 14/17 tested

were independently validated. Deep conservation analysis of all sORFs in *C. jejuni* and *C. coli* lineages with diverse lifestyles revealed highly conserved CioY (34 aa), which we show is a small component of the CioAB terminal oxidase. Our translatome data and annotation refinements are available in our interactive *CampyBrowse* resource at Ribo-base (http://www.bioinf.uni-freiburg.de/ribobase). Overall, we describe a generic strategy for approaching translatome refinement and small protein discovery in prokaryotes and provide a resource highlighting new small genes and translatome features that might affect *C. jejuni* physiology.

## Results

### Ribo-seq in *C. jejuni* distinguishes between coding and non-coding regions

To globally identify and map translated *C. jejuni* ORFs, we established three complementary Ribo-seq approaches for the widely-used lab strain NCTC11168 grown under standard conditions (log-phase growth in rich media) in parallel with RNA-seq to measure the transcriptome (Fig. 1a). In addition to canonical Ribo-seq, we applied TIS profiling using Ret ("Ribo-RET") and Onc treatment to map start codons[20,22]. Furthermore, we generated a de novo dataset to detect TTS (stop codons) based on Ribo-seq analyses after apidaecin-137 (Api) treatment, as shown to enrich ribosomes at stop codons only in *E. coli*[23,29]. The analysis and re-annotation of the *C. jejuni* coding genome using these datasets is described in the subsequent sections of this study. The generated data and refined translatome annotations are provided in our above-listed interactive *CampyBrowse* browser.

We first established 'standard' Ribo-seq experimental and data analysis protocols (with Cm[30]) for *C. jejuni* (Supplementary Fig. 1a, see **Methods**). In general, our dataset successfully differentiated between known coding and non-coding regions. For example, cDNA reads for the sRNA CJnc180[24,25] were mainly restricted to the parallel transcriptome library, while the adjacent ORF Cj1650 had both RNA-seq and Ribo-seq coverage (Fig. 1b). Housekeeping ncRNAs (hkRNAs) likewise showed low footprint coverage (Supplementary Fig. 1b).

A global examination of translational efficiencies (TE: ratio of coverage in Ribo-seq/RNA-seq libraries; ribosome occupancy normalized to transcript abundance) also showed a clear difference between coding and non-coding features, with ORFs having a mean TE ≥ 1 and tRNAs/hkRNAs < 1 (Fig. 1c). The TE of mRNA leaders was on average higher than coding regions. This might reflect the short length of *C. jejuni* 5'UTRs (~ 25 nt), which means a significant length is protected by initiating ribosomes[24,31]. Also, re-initiation can occur in the absence of elongation, especially with Cm treatment, which might further enrich coverage in 5'UTRs adjacent to start codons[32]. Nonetheless, long annotated 5'UTRs lacked reads in Ribo-seq but not RNA-seq libraries (Supplementary Fig. 1c). While most intergenic sRNAs had a TE < 1, indicating they are in fact non-coding, those overlapping ORFs were enriched in Ribo-seq (e.g., CJnc11; TE: 1.94). However, two intergenic sRNAs had a TE ≥ 1 (CJnc60 & CJnc110; 1.54 & 2.83, respectively), suggesting they might be small mRNAs or dual-function sRNAs encoding an sORF (Fig. 1c). In line with this, it has been proposed that CJnc60 encodes an unannotated SelW homolog (81 aa)[33]. Based on this successful set-up of Ribo-seq in *C. jejuni*, we next inspected the translation of annotated small proteins in more detail.

### Examining the translation status of the annotated *C. jejuni* small proteome

The *C. jejuni* NCTC11168 annotation (from 2021-09-11) includes 54 small proteins (here based on a definition of ≤ 70 aa) and none below 30 aa (Supplementary Data File 1). These 54 small proteins include ten small ribosomal proteins (r-proteins) and a handful with housekeeping functions (e.g., Dba, SecE), but mostly hypothetical, unvalidated proteins (Fig. 1d and Supplementary Data file 1). Most had a TE > 1 (mean ~ 4) in our dataset, supporting that they are indeed translated (Fig. 1c).

To confirm their translation, as well as to guide discovery of novel sORFs not already annotated in *C. jejuni* NCTC11168 (hereafter, "novel") based on our Ribo-seq data (below), we next generated C-terminal 3xFLAG or SPA (Sequential Peptide Affinity) epitope-tagged versions of 41 out of 44 annotated non-ribosomal sORFs. We also included Cj0185c (69 aa), which had been removed from the NCBI annotation (2021-09-11) but caught our eye during initial inspection of Ribo-seq data (Supplementary Fig. 1d). For three out of these 45 sORFs, we could

not generate a tagged version at the native locus, suggesting that the epitope interfered with an essential function (Cj1047c/63 aa, Cj1160c/59 aa, *secE*/59 aa).

Western blotting detected 35/42 tagged small proteins in at least one condition, and 34 in the same growth phase as Ribo-seq (log) (Fig. 1e, Supplementary Fig. 1e and Supplementary Data file 1). Although most showed stable or decreased levels at later growth phases, some accumulated at least two-fold in the stationary phase,

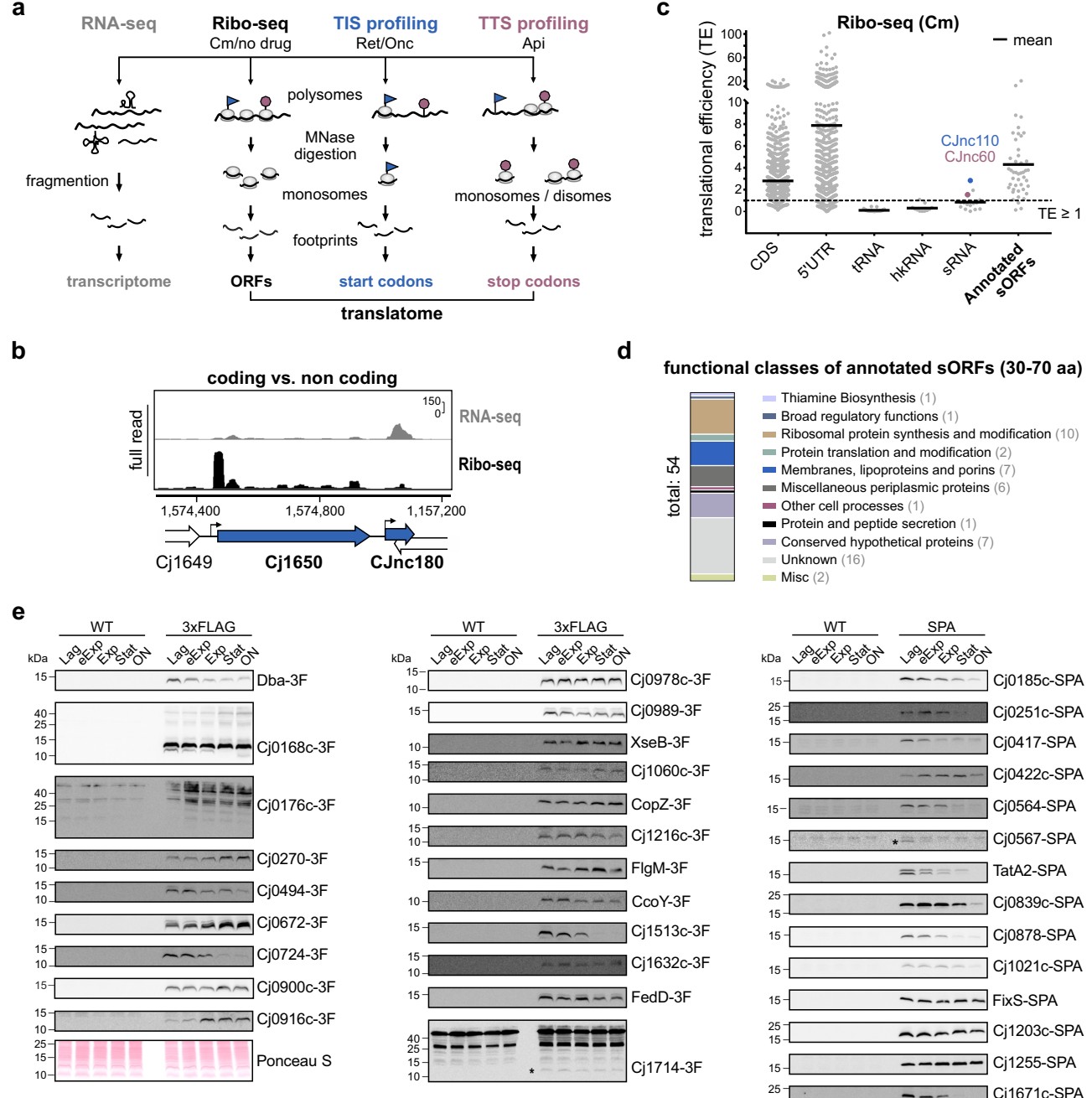

**Fig. 1 | Establishing Ribo-seq in *C. jejuni* with the annotated small proteome.**
**a** Overview of Ribo-seq techniques applied/developed in this study. MNase: micrococcal nuclease, Cm: chloramphenicol, Onc: oncocin, Ret: retapamulin, Api: apidaecin. **b** Ribo-seq and paired RNA-seq distinguish the coding gene Cj1650 from the non-coding sRNA CJnc180[25]. *Y*-axis: rpm (reads per million). Representative of *n* = 3 independent experiments. **c** Translational efficiency (TE: Ribo-seq/RNA-seq) from Cm-treated Ribo-seq for different feature classes in the *C. jejuni* annotation[24,44]. A total RNA RPKM of ≥30 was required. CDS: coding sequences/

ORFs. hkRNA: housekeeping RNAs. **d** Functional classes of annotated sORFs[44]. See also Supplementary Data file 1. **e** Translation validation by western blot for annotated sORFs with C-terminal 3 × FLAG (3F) or SPA epitope tags with an anti-FLAG antibody. Ponceau S staining of membranes was used as loading control. eExp: early exponential. Stat: Stationary. ON: overnight. Untagged WT: antibody (anti-FLAG) control. Representative of *n* = 2 independent experiments. Asterisks: low abundance FLAG-specific bands. Source data are provided as a Source Data file.

suggesting they might mediate adaptation under these conditions (Cj0672, Cj0916c, CopZ, Fig. 1e). MS analysis of whole cell protein samples also validated translation of 15/55 sORFs, including six small ribosomal proteins, as well as Cj0185c and untaggable Cj1047c (Supplementary Data File 2). Two or more unique peptides were detected for the majority of these.

Overall, our Ribo-seq and complementary western blot/MS analysis suggest that most annotated *C. jejuni* sORFs are translated in the log phase. This analysis also provides a benchmark set to guide novel sORF detection by Ribo-seq, TIS, and TTS (below).

### Retapamulin-based translation initiation site (TIS) profiling reveals start codons

Assigning Ribo-seq coverage to sORFs that overlap longer ORFs can be challenging. To increase confidence in detecting translation in these complex genomic contexts, we next established TIS profiling in *C. jejuni* to reveal start codons (Fig. 1a). A Δ*tolC* mutant strain was used previously in *E. coli* for Ribo-RET to reduce retapamulin efflux[20]. Deletion of the gene encoding the major efflux component CmeB in *C. jejuni*[34] reduced the Ret minimum inhibitory concentration (MIC) 8-fold without an effect on growth (see **Supplementary Methods**; Supplementary Fig. 2a). Ret treatment for 10 min of the Δ*cmeB* mutant also collapsed polysomes to monosomes even without micrococcal nuclease (MNase) digestion (Supplementary Fig. 2b), as in *E. coli*[20], indicating successful stalling of initiating ribosomes and run-off of elongating ribosomes. We also omitted Cm treatment of the control culture to avoid potential bias in overall Ribo-seq coverage[32] and instead harvested cells by fast-filtration (see **Methods**), which still recovered polysomes and led to the enrichment of ORFs, but not ncRNAs (Supplementary Fig. 2b, c).

To determine whether Ret treatment in *C. jejuni* successfully enriched ribosome occupancy at start codons genome-wide, we performed metagene analysis of cDNA coverage near all annotated start codons (ATG, TTG, GTG). This showed enrichment in the Ret vs. no drug libraries at −16/+16 nt upstream/downstream of start codons for 5′/3′ read end coverage, respectively (Fig. 2a and Supplementary Fig. 2d, e). As reported for MNase in other bacteria, trimming at 3′ ends was sharper than at 5′ ends[32] (Supplementary Fig. 2d, e). In addition to using 3′ end mapping, we selected read lengths giving the best enrichment in metagene analysis to further avoid blurring of P-site offsets due to imprecise MNase trimming as performed previously[32,35]. Our protocol recovers RNAs between 26-34 nt (Supplementary Fig. 2f)[36], but enrichment at TIS was strongest for 31/32 nt-long reads.

Manual inspection of annotated ORFs using these optimal read lengths and offsets revealed TIS signals at the expected position. For instance, at a highly translated r-protein operon, we detected enrichment ~16 nt downstream of each start codon (Supplementary Fig. 3a). We also observed strongly enriched TIS peaks for three previously predicted but unannotated *C. jejuni* leader peptides (LeuL, TrpL, and MetL)[31] (Fig. 2b and Supplementary Fig. 3b). Ret treatment also revealed a putative internal TIS in *lysC* driving translation of a LysCβ subunit, in agreement with the short isoform previously reported in *B. subtilis* (Supplementary Fig. 3c)[37].

About 20 leaderless mRNAs have been predicted in *C. jejuni*[24,31]. Our size selection during cDNA library preparation omits the short 15 nt footprints that would be expected for ribosomes initiating at leaderless start codons. Thus, we inspected these candidates with both 5′ and 3′ mapping for signals at the start codon/30 nt downstream, respectively, as well as differential RNA-seq (dRNA-seq) data, as done previously[14]. This, together with the absence of any in-frame start codon in the first 30 nt, suggested that six genes initiate translation at or near the TSS (Supplementary Data File 3). This includes Cj0667 (Fig. 2c), which was also reported to be leaderless in *Helicobacter pylori* 26695[38]. In contrast, our TIS data suggest that Cj0459c is leadered and requires start codon re-annotation (Supplementary Fig. 3d). The status

of 14 predictions remains unclear, due to low coverage and/or absence of TIS peaks (Supplementary Data File 3). Therefore, there might be additional leaderless ORFs, which could be captured by modified Ribo-seq approaches that retain shorter read lengths (including the expected ~15 nt footprints of leaderless transcripts[13]) or an independent method. Overall, metagene analysis and single-gene inspection supported the successful establishment of TIS profiling and its potential to map start codons in *C. jejuni*.

### Translation initiation site signals from Ret treatment refine start codons

Accurate start codon information is crucial for, e.g., studying post-transcriptional regulation at 5′UTRs, but manually annotating them can be laborious. For automated detection of start codons based on TIS data, we adapted an approach for genome-wide detection of TIS peaks (see **Methods** for details)[13]. Using the above-mentioned read-lengths and offsets from metagene analysis (Supplementary Fig. 2e), we detected approx. 50% of annotated start codons in our TIS(Ret) dataset (Supplementary Fig. 3e). Further inspection of those without TIS signal showed that 20 had TIS peaks fitting in-frame start codons within 51 nt of the annotated start, strongly suggesting that they might require re-annotation. Thus, we inspected Ribo-seq/TIS data for these 20 ORFs as well as 115 ORFs previously marked for re-annotation based on their TSS position in four *C. jejuni* strains[24]. In total, we re-annotated start codons for 28 out of 1576 genes in strain NCTC11168 (mostly shortening) supported by conservation[24] (Fig. 2d and Supplementary Data Files 4, 5). Moreover, 26/28 re-annotated ORFs had a predicted RBS motif after, but not before, re-annotation (Fig. 2d, *right*). However, 40% of inspected genes could not be clearly annotated due to low sequencing coverage.

The 28 re-annotated genes included sORF Cj0900c, where the new start codon generates an even shorter protein (48 vs. 59 aa) (Fig. 2e). Several relatively well-characterized proteins also have shorter N-termini than annotated, including FliA (motility sigma factor, Supplementary Data File 5) and NrfH (nitrite reductase small subunit) (Fig. 2f and Supplementary Data File 5). While re-annotating the Cj0055c start codon, we also identified Cj0056c, which, like Cj0185c, should be added back to the NCBI annotation (Supplementary Fig. 1d and Supplementary Data File 5). MS data revealed two additional N-terminal extensions (Cj0636 and Cj1253 (*pnp*)) (Supplementary Data Files 2, 5).

Altogether, these observations suggest the successful establishment of TIS profiling with Ret in *C. jejuni*. Our data allowed us to detect previously-predicted leader peptides, to inspect several predicted leaderless mRNAs, and to refine start codon coordinates for 28 ORFs. This shows the overall utility of TIS profiling for annotation of *C. jejuni* coding regions.

### Apidaecin treatment enriches for terminating monosome and disome footprints

While internal TIS can generate shorter, functional proteins[4], C-terminal truncations that arise via, e.g., point mutations, alternative decoding, or ribosomal frameshifting are less well characterized. These features are not easily detected from the reference genome. Detection of stop codons (TTS) might also provide additional evidence for canonical (s)ORF translation. Thus, we established a Ribo-seq-based TTS profiling method to globally map stop codons, which uses the PrAMP Api to trap terminating ribosomes[29,39] and enrich Ribo-seq coverage at stop codons as applied previously to *E. coli*[23,29,39] (Fig. 1a).

Although some *C. jejuni* strains are not sensitive to natural api-daecin from honeybees[40], our NCTC11168 WT isolate had a minimum inhibitory concentration (MIC) for synthetic apidaecin 137 (hereafter, Api) of ~6.25 μM, which was dependent on Cj0182, encoding a protein with homology to the SbmA peptide transporter required for uptake of PrAMPs in *E. coli*[22] (see **Supplementary Methods**). We treated a *C. jejuni* WT culture with ~10 × MIC Api (50 μM) for 10 min. We also

treated the culture with the same concentration of Onc (MIC ~ 6.25 μM) for parallel TIS profiling, as *C. jejuni* WT is not sensitive enough to use Ret. Like Ret, Onc collapsed polysomes into monosomes without MNase treatment (Fig. 3a and Supplementary Fig. 2b). In contrast,

polysomes from *C. jejuni* treated with Api showed MNase resistance (Fig. 3a, *right*). We hypothesize that ribosome queuing at the stop codon in Api-treated cells interferes with MNase access. This has been reported in eukaryotic ribosomes near stop codons and collision sites

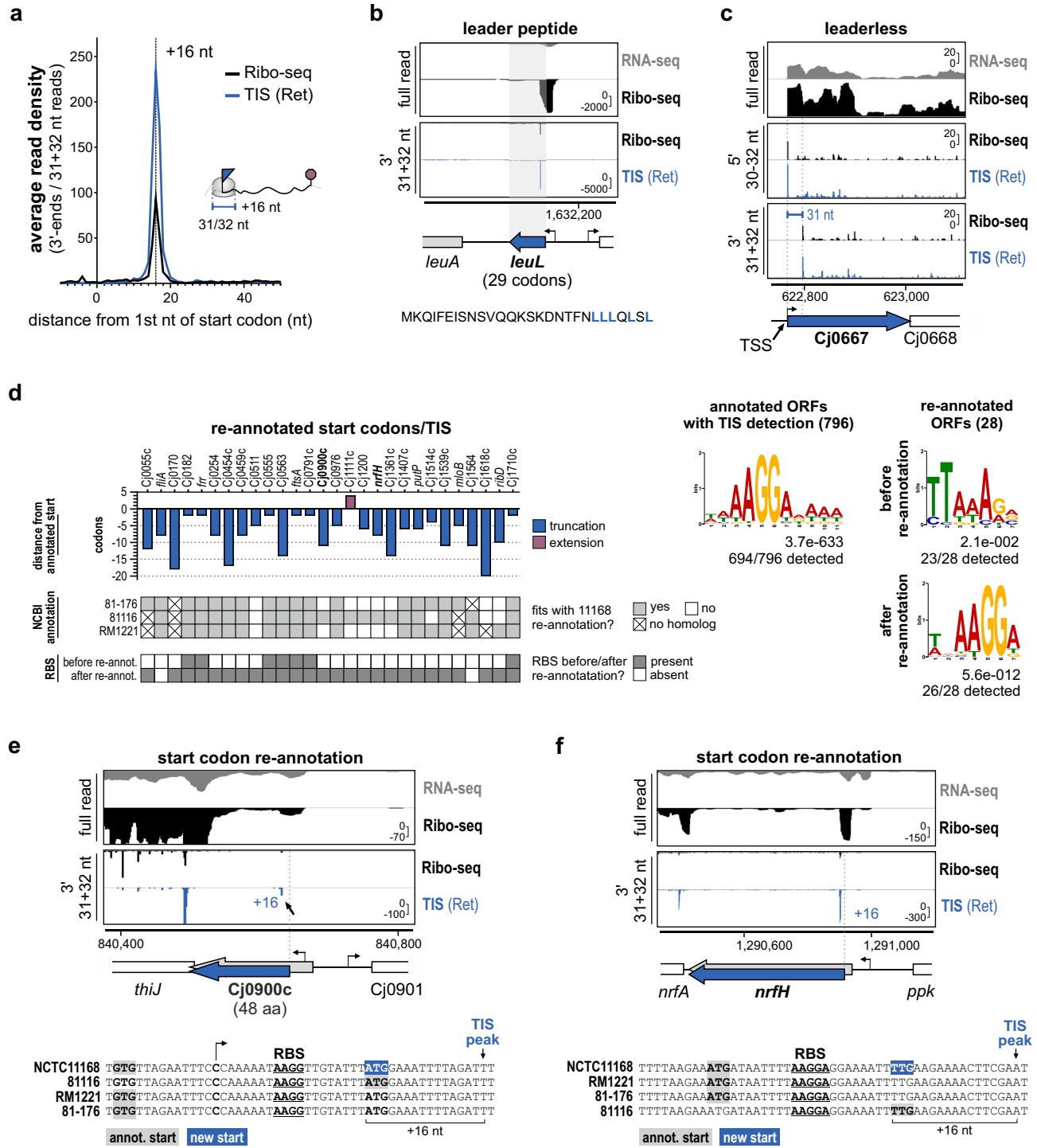

**Fig. 2 | Ribo-seq & TIS-based mapping of start codons. a** Genome-wide ribosome occupancy at start codons for no drug (Ribo-seq) and TIS(Ret) libraries. Representative of *n* = 3 independent experiments. See also Supplementary Fig. 2d, e. **b** LeuL leader peptide (CJsORF41, Supplementary Data file 1). *Below*: proposed amino acid sequence[31]. **c** Leaderless ORF Cj0667[31]. Leaderless ORFs are expected to have 5'/3' peaks at the TSS/30 nt downstream of the TSS, respectively. **d** Start codon re-annotations based on Ribo-seq and TIS (see also Supplementary Data file 4). *Right side:* RBS motif predictions. *Left*: *C. jejuni* consensus RBS motif identified in 15 nt upstream of all annotated ORFs with a detected TIS. *Right*: RBS motif prediction

based on annotated start codon (before) and with new re-annotated start codon (after). Motifs were predicted with MEME[85]. **e** Re-annotation of Cj0900c sORF start codon. Gray: Annotation (59 aa)[44]. Blue: Re-annotation (48 aa). **f** Re-annotation of the *nrfH* start codon. *Bottom*: alignment of translation initiation region from four *C. jejuni* strains. Gray - annotated start codons. Blue - new NCTC11168 start codon. Underlined - potential RBS. Bent arrows - TSS[24]. Arrow - Ret-enriched peak. For screenshots, *y*-axis: rpm (reads per million); representative of *n* = 3 independent experiments. Source data are provided as a Source Data file.

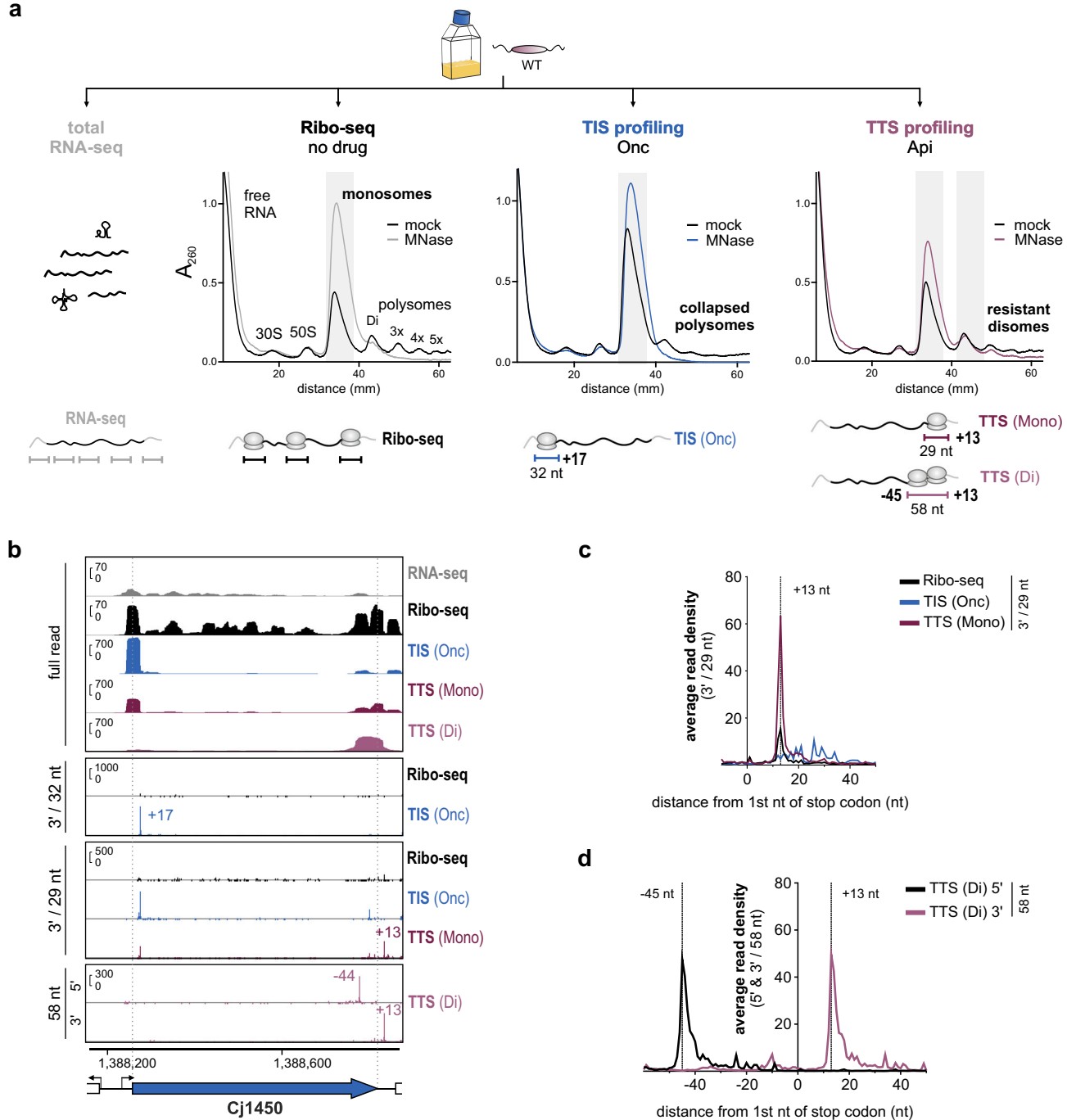

**Fig. 3 | Apidaecin treatment combined with Ribo-seq for mapping translation termination sites. a** Sucrose density gradient analysis of ribosome species in lysates for the TTS experiment for untreated (no drug), Onc (TIS), or Api (TTS) treated *C. jejuni* WT cultures. Shaded regions: fractions harvested for library preparation. Representative of *n* = 3 independent experiments. *Below:* Overview of libraries, footprint lengths, and 5′/3′ peak offsets used for TIS/TTS automated detection are shown (see also panels C/D). **b** Ribo-seq cDNA coverage (full-read and single-nt) for Cj1450. Single-nt coverage files (first or last base of reads) were generated with read lengths giving the strongest/sharpest enrichment/peaks (see panels C/D). Y-axis: rpm (reads per million). Representative of *n* = 3 independent experiments. **c, d** Metagene analysis of ribosome occupancy at annotated stop codons for 3′ ends of 29 nt reads for the monosome library or 5′/3′ ends of 58 nt reads for the disome library. All read lengths: Supplementary Fig. 4. −45 nt/+13 nt: offsets used for TTS(Di) and (Mono) peaks, respectively. Source data are provided as a Source Data file.

and is exploited for 'disome' profiling[41–43]. We thus also prepared libraries from disome footprints (50-80 nt) for Api-treated cultures. In total, our TTS profiling experiment included five libraries: RNA-seq, untreated Ribo-seq, TIS profiling using oncocin [TIS(Onc)], and two TTS profiling libraries: monosome [TTS(Mono)] and disome [TTS(Di)] (Fig. 3a, *bottom*; for read length distribution see Supplementary Fig. 4a).

Full-read coverage plots for single ORFs revealed enrichment of coverage towards the 3′ end for both the TTS(Mono) and TTS(Di) libraries compared to no-drug or TIS(Onc), suggesting that Api was stalling terminating ribosomes and inducing queuing (Fig. 3b). Metagene analysis of 5′/3′ read end coverage near annotated stop codons genome-wide showed enrichment at −16 nt/+13 nt with respect to the stop codon in the TTS(Mono) vs. no drug libraries (Fig. 3c). This was

most pronounced for 29 nt reads (Supplementary Fig. 4b). Enrichment of 3' end coverage by Onc at +17 nt vs. start codons contrasted with enrichment at +13 nt vs. stop codons for Api (Supplementary Fig. 4b–d). This 3 nt difference is in line with observations in *E. coli*, which show that Api leaves the stop codon in the A-site, while initiation inhibitors stall the ribosome with the start codon in the P-site[20,39]. For TTS(Di) libraries, peaks of ribosome occupancy near stop codons were highest at −45/+13 nt for 5′/3′ end mapping, respectively, for 58 nt reads (Fig. 3d and Supplementary Fig. 4e).

To determine if our data could recover stop codons, we first manually inspected read-length optimized coverage files for a highly translated ribosomal protein-encoding (r-protein) operon. This showed strongly enriched peaks in the TTS(Mono) vs. Ribo-seq library at the expected position downstream of each stop codon (Supplementary Fig. 5a). TTS(Di) coverage also showed peaks at positions in line with metagene analysis (Supplementary Figs. 4e, 5a). We observed some artifacts of Api treatment previously observed in *E. coli*[29], including increased coverage in some 3′UTRs, presumably caused by readthrough of stop codons for both TSS(Mono) and TTS(Di) libraries (Supplementary Fig. 5b). We also observed enrichment vs. normal Ribo-seq at some start codons in TTS(Mono) libraries (Fig. 3b and Supplementary Fig. 4c). Our parallel Onc library, therefore, served not only to map start codons in WT, but also to control for this artifact. Start codon enrichment was also less pronounced for the TTS(Di) compared to the TTS(Mono) library, suggesting disome profiling might also circumvent some artifacts induced by Api. Overall, this suggests that despite some complexity around stop codons, Api treatment paired with Ribo-seq reveals sites of translation termination.

### TTS profiling reveals stop codon usage not apparent from the reference genome

To further demonstrate the utility of Api, we inspected several previously described/predicted *C. jejuni* genes that have TTS that would not be immediately apparent from the reference genome. This included those with recoding, pseudogenes, and phase-variable genes generated by single-point mutations in coding regions. For example, predicted *selW* ORF on CJnc60 requires decoding of a specific UGA stop codon by tRNA-SeC to incorporate a selenocysteine[33]. In line with the full translation of 80 aa SelW, Ribo-seq revealed ribosome occupancy across the CJnc60 transcript, and a TTS peak was present after the putative full-length *selW* stop codon (Fig. 4a). Proteomics further confirmed full-length translation of SelW beyond the Sec insertion site (Supplementary Data File 2).

We next inspected a list of pseudogenes[44] generated by single nucleotide changes (via strain variation or sequencing errors). Consistent with their genotype, several showed coverage only until the annotated premature stop codon as well as an associated TTS peak, in some cases generating novel sORFs (Supplementary Fig. 5c, d). In contrast, coverage for others appeared different from what would be expected from their annotation. For instance, Ribo-seq coverage for pseudogene Cj0455c extended beyond an annotated, premature TAA stop codon, and a TTS peak was detected downstream of the potential full-length ORF stop codon (Fig. 4b). In line with this, a 61 aa C- terminal extension was previously reported in some *C. jejuni* NCTC11168 isolates[45], and Sanger sequencing of our strain showed a similar TAA → CAA conversion (Fig. 4b). To demonstrate that full-length Cj0455c is translated, we fused a SPA epitope to the penultimate codon of the ORF at the native locus. Western blot analysis detected the translation of a protein of the expected full-length size (20.6 kDa + SPA tag) (Fig. 4b, *bottom*). Nevertheless, we cannot fully exclude a heterogeneous population with some bacteria carrying the stop codon mutation, as a putative TTS peak also fits the premature stop codon. Cj0455c is required for *C. jejuni* motility, a key virulence feature[46]. We also deleted Cj0455c, which resulted in a non-motile phenotype. Introduction of a full-length (CAA), but not a truncated

(TAA), copy of Cj0455c into ΔCj0455c at the unrelated *rdxA* locus partially restored motility (Fig. 4b, *bottom right*; Supplementary Fig. 6a). Altogether, this suggests Cj0455c should be re-annotated as an intact motility-associated ORF.

### TTS-profiling detects the ON/OFF status of phase-variable genes

*C. jejuni* regulates several ORFs via ON/OFF phase variation due to length variation of hypermutable homopolymeric tracts (e.g., polyG, polyA)[28,47]. Several genes with sufficient read coverage had TTS peaks in line with their reference genome homopolymeric tract length. However, coverage for two examples, previously found to be variable in human challenge experiments[48], did not match their genotype. Reads for Cj0170 (reference G8-ON) covered only the first half of the ORF until just after the G-tract, and a TTS(Di) peak was also visible just downstream of the polyG sequence (Fig. 4c, *left*). This is in line with a premature in-frame stop codon due to a G9 tract, which we confirmed by Sanger sequencing of our isolate. Paired TIS(Onc), data also showed that the Cj0170 start codon should be re-annotated downstream of its TSS, as suggested previously[24] (Supplementary Data File 4, 5). Coverage for a second phase-variable ORF, Cj1325, was also inconsistent with its reference G-tract length (G10-OFF), as it extended across the entire ORF. Peaks were also detected in the Api-treated libraries immediately downstream of the full-length stop codon (Fig. 4c, *right*). Consistent with Ribo-seq and TTS, Sanger sequencing showed that our isolate carried a G9-ON sequence.

*C. jejuni*'s characteristic helical cell shape is also under phase variable control. A common change associated with rod morphology is a single nt conversion in a mutable A-stretch (A8-A7) in *pgp1* (Cj1345c, peptidoglycan endopeptidase)[49]. The A7 stretch generates a premature stop codon that truncates 14% of the C-terminus of the protein to render it non-functional (Supplementary Fig. 6b). Our NCTC11168 isolate displays straight morphology. TTS peaks of *pgp1* fit the premature in-frame stop codon for the A7-OFF allele (Fig. 4d (*top*); Supplementary Fig. 6c). Sanger sequencing confirmed that our isolate harbors an A7 tract. Finally, to demonstrate again that TTS profiling can reveal physiologically relevant genomic changes, we added an A8-ON allele of *pgp1* (from the spiral strain 81-176) to our NCTC11168 WT isolate at the unrelated *rdxA* locus. This restored spiral morphology (Fig. 4d (*bottom*); Supplementary Fig. 6d). Altogether, inspection of *C. jejuni* ORFs that frequently change their translation status due to single nucleotide variations from the reference sequence demonstrated the potential of TTS profiling to inform on hidden translation events.

### Discovery of novel sORFs in diverse genomic contexts

Finally, we generated novel sORF predictions from all of our datasets. For the 'standard' Ribo-seq, we used two published tools in our HRIBO pipeline: REPARATION and DeepRibo[50–52]. For TIS-based predictions, our above-mentioned peak detection approach was used, which was also used with minor modifications to predict sORFs based on TTS peaks (see **Methods**). As these tools can generate long lists of candidates with little guidance for ranking and many are likely false positives[53], we used our western blot/MS-validated small proteins (Fig. 1e; Supplementary Data file 1) to set cutoffs for RNA-seq RPKM, TE, the score generated by DeepRibo, and peak height/enrichment (Supplementary Fig. 7a, b, see **Methods** for details). With relatively stringent cutoffs, Ribo-seq, TIS, and TTS-based predictions recovered 39, 30, 33/45 validated sORFs, respectively (Supplementary Data File 1). Of the 33 TTS-detected benchmark sORFs, only 10 were detected with both monosome and disome peaks, highlighting the complementarity of these two approaches (Supplementary Fig. 7c). Most ncRNAs (72/86) were not detected, other than abundant housekeeping RNAs (e.g., tRNAs, tmRNA, 6S RNA, RnpB) and three potential dual-function sRNAs/small mRNAs CJnc60 (SelW, Fig. 4a), CJnc110, and CJnc190 (Supplementary Data File 6). Overall, our systematically validated annotated sORF set indicated that the tools can identify

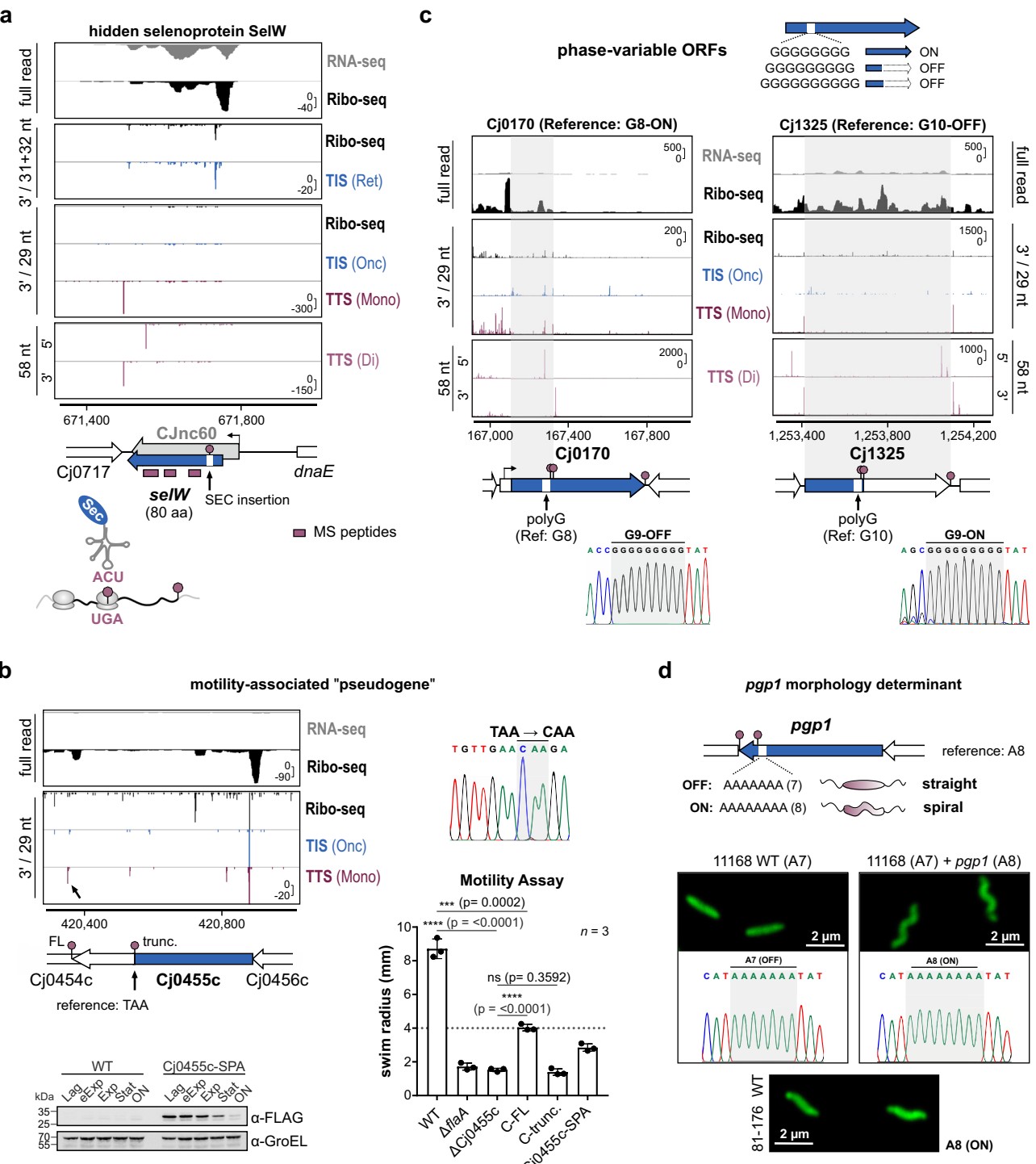

**Fig. 4 | TTS profiling reveals translation status of infection-relevant genes not apparent from the reference genome. a** Translation of the selenoprotein *selW*. Gray: CJnc60 sRNA[24]. Blue: *selW* ORF[33]. Red: detected MS peptides. Stop sign: UGA read by tRNA-Sec. **b** Full-length (FL) translation of motility-related "pseudogene" Cj0455c. *Top left*: Genomic context with premature TAA stop codon. TTS(Mono) peak (arrow) for FL Cj0455c. (*Top right*) Sanger sequencing of TAA-CAA conversion. (*Bottom left*) Western blot validation of FL translation (anti-FLAG). Representative of *n* = 2 independent experiments. *Bottom right*: Motility in soft agar of ΔCj0455c complemented either with FL (C-FL; CAA) or truncated (C-trunc.; TAA) Cj0455c isoform at *rdxA*. The mean/standard deviation of swim radii for three biological replicates is shown. Δ*flaA*: non-motile control. ****: *p* < 0.0001, ***: *p* < 0.001, ns: *p* > 0.05 (not significant), two-sided, unpaired Student's *t* test. ΔCj0455c vs. WT:

<0.0001; C-FL vs. ΔCj0455c: < 0.0001; C-trunc. vs. ΔCj0455c: 0.3592; C-FL vs. WT: 0.0002. See also Supplementary Fig. 6a. **c** TTS peaks for phase variable genes Cj0170 and Cj1325. Blue arrows: current NCBI gene annotation. Gray shading: Translated region of ORFs. *Below*: Sanger chromatograms of G-tract in WT used for Ribo-seq/TTS. **d** *pgp1* TTS peaks, genotype, and morphology of WT isolate used for Ribo-seq. *Top*: the genomic context of *pgp1* A-tract. A8: ON, A7: OFF/truncated. *Bottom*: Morphology of rod-shaped WT supplemented with *pgp1*-A8(ON) allele. Cells were stained with FITC (fluorescein isothiocyanate) and imaged by confocal microscopy. Representative of *n* = 2 independent experiments. See also Supplementary Fig. 6b–d. *Y*-axis: rpm (reads per million). Representative of *n* = 3. Source data are provided as a Source Data file.

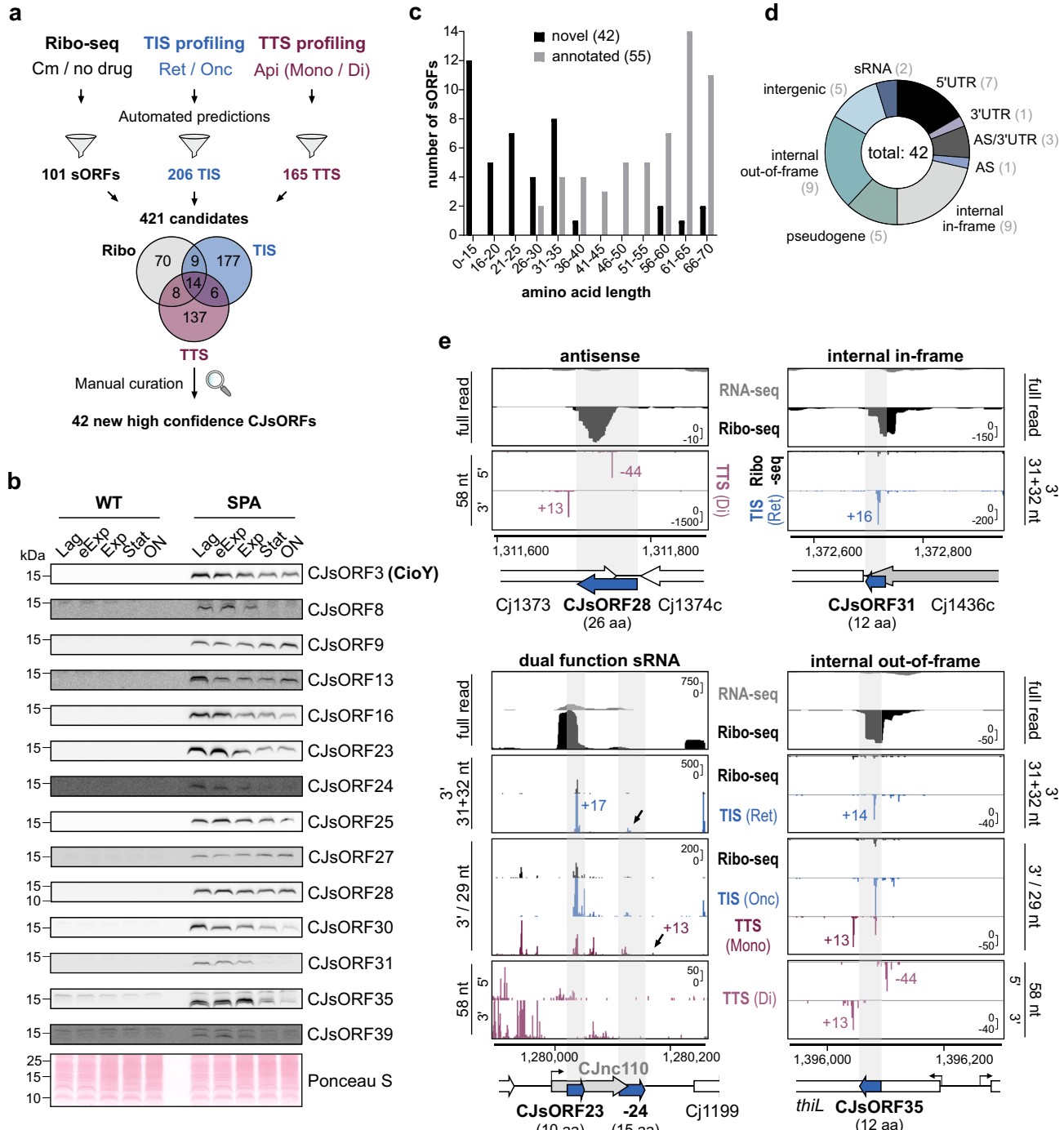

**Fig. 5 | Novel *C. jejuni* sORFs revealed by integrated translatomics. a** Workflow for expanding the *C. jejuni* small proteome via automated predictions based on ribosome occupancy, TIS, and TTS. Predictions were filtered (see **Methods;** Supplementary Fig. 7d) for a set of 421 strong candidates, which were manually inspected for Ribo-seq, TIS, and TTS coverage & RBS. **b** Western blot validation of selected novel CJsORFs with a C-terminal SPA tag on Tricine-SDS-PAGE gels. SPA-tagged proteins were detected with an anti-FLAG antibody. Ponceau S staining of membranes serves as a loading control and is shown only once. Representative of *n* = 2 independent experiments. eExp: early exponential. Exp: exponential. Stat: Stationary. ON: overnight. **c** Length distribution of novel vs. annotated CJsORFs. **d** Genomic context of novel CJsORFs. AS: antisense. **e** Coverage for selected validated novel CJsORFs in diverse contexts by Ribo-seq, TIS, and/or TTS. See also Supplementary Fig. 10. *Y*-axis: rpm (reads per million). Representative of *n* = 3. Source data are provided as a Source Data file.

translated sORFs from our data with both sensitivity and specificity (Fig. 1e and Supplementary Data File 1), although not all Ribo-seq methods detected each sORF.

With the parameters described above, we compiled a filtered list of 421 sORF candidates (11-71 codons) from the predictions generated from our three Ribo-seq datasets (Fig. 5a) (see **Methods;** Supplementary Fig. 7d for filtering details). In addition, the top 40 TIS candidates

below 10 codons were also included. The candidate list from automated predictions included, e.g., the above-mentioned leader peptides *leuL, trpL,* and *metL* (Fig. 2b and Supplementary Fig. 3b) or the re-annotated version of Cj0900c (Fig. 2e), supporting our filtering approach.

We next manually curated the automated predictions in a genome browser based on their Ribo-seq, TIS, and TTS coverage (Supplementary Data File 6). We considered the shape of Ribo-seq coverage with

respect to the sORF prediction, TIS/TTS peak position and height with respect to the prediction start/stop codon, and enrichment (e.g., RNA-seq vs. Ribo-seq, TIS vs. Ribo-seq), as well as the presence of a potential RBS or transcriptional start site (for details see **Supplementary Methods &** Supplementary Table 1). This left us with a list of 42 new high-confidence "CJsORFs" ≥ 5 codons, numbered based on genome position in strain NCTC11168 (Supplementary Data File 1). In addition, we identified 18 putative start-stop sites (2 or 3 codons, internal out-of-frame, Supplementary Data file 7). Such start-stop sites were previously shown to have potential regulatory functions in *E. coli*[20]. None of the CJsORFs were detected in our parallel MS-based proteomics survey using small protein-targeted approaches (Supplementary Data File 2). However, 37/42 had an RBS-like sequence, supporting that they are true sORFs (Supplementary Fig. 8a). We selected 17 from diverse genomic contexts for validation by SPA tagging. Western blot analysis showed that 14 are robustly translated in at least one growth phase in a rich medium (Fig. 5b and Supplementary Fig. 8b, c). Low levels of three potential internal in-frame candidates (CJsORF7, CJsORF18, CJsORF36) were also detected (Supplementary Fig. S8d–f). We cannot, however, rule out that detected bands represent C-terminal fragments of the parental protein generated by proteolysis, which retains the epitope tag and leave these candidates as so far unvalidated. While many were constitutively expressed, others accumulated or decreased at least ~ 2-fold between exponential and stationary phase (e.g., CJsORF9 (33 aa) or CJsORF3 (34 aa), respectively, Fig. 5b). We also tagged three candidates that were included in the filtered list of the 421 sORF predictions, but discarded during manual curation. None of the three could be detected on a western blot, supporting our manual curation approach (Supplementary Fig. 9a–c). However, we cannot rule out that these three sORFs, and even additional ones not in our list based on log phase in rich media, might be translated under other conditions.

The 42 novel CJsORFs were, in general, shorter than those in the annotation, including 28 with a length of ≤ 30 codons and the shortest encoding a protein of only 4 aa in length, excluding the putative start-stop sites (Fig. 5c and Supplementary Data File 7). This included CJsORF19 (6 aa), encoded upstream of Cj0772c (Supplementary Fig. 10a). The new CJsORFs were encoded in diverse genomic contexts (Fig. 5d). For example, we validated a 26 aa sORF (CJsORF28) in the 3' UTR of Cj1374c (purine NTP pyrophosphatase) overlapping the Cj1373 membrane protein gene on the antisense strand (Fig. 5e (*top left*); Supplementary Fig. 10b). Our data also revealed two sORFs (CJsORF23 and CJsORF24; 10 and 15 aa, respectively) on CJnc110 sRNA, which we could validate by western blot (Fig. 5b, e, *bottom left*). TIS profiling revealed CJsORF31 translated from an internal in-frame start codon in Cj1436c and comprising only the last 12 aa of the 390 aa parental ORF (Fig. 5e (*top right*); Supplementary Fig. 10c). Finally, all three Ribo-seq methods revealed even internal out-of-frame examples, such as CJsORF35 (12 aa) within *thiL*, encoding thiamine monophosphate kinase (Fig. 5e, *bottom right*).

Altogether, our translatomics approach, supported by independent western blot validation, expanded the number of *C. jejuni* small proteins almost by a factor of two (from 55 to 97) and suggested that the components of the *C. jejuni* small proteome are even shorter than what was previously annotated.

## Comparative genomics reveals conserved and strain-specific sORFs

To explore potential functions for the 97 sORFs (annotated and novel), we inspected their conservation and primary sequence features. First, we used tBLASTn to search for potential matches in public Epsilon-proteobacteria genomes (Supplementary Data File 8) with parameters used previously for bacterial sORFs[54]. As expected, all 10 small ribo-somal proteins were highly conserved, as well as several with known functions (e.g., SecE, FlgM (Supplementary Fig. 11a and summarized in Fig. 6a). Potential leader peptides (*trpL, leuL, metL*) were generally

detected only in *C. jejuni* or related *Campylobacter* species. Several with only hypothetical functions are highly conserved, including Cj0270 (tautomerase), Cj0916c (selenoprotein[55]), and previously-missing Cj0185c (PhnA-domain). Others appear to be strain- or species-specific (e.g., putative transcriptional regulator Cj0422c, absent in widely-studied strain 81-176; Cj0168c, absent outside of *C. jejuni*). While many highly conserved CJsORFs were generally those internal in-frame in annotated proteins with core functions (e.g., CJsORF18, 20 aa, in *flgH*) (Fig. 6a and Supplementary Fig. 11a), two (CJsORF3, 3'UTR of *cydB* oxidase component (34 aa) and CJsORF25, 5'UTR of *guaA* GMP synthase (69 aa)) were also detected in most *Campylobacter* species.

To provide an expanded conservation analysis, we also inspected sORF presence vs. lineage/isolation source in the thousands of available genome sequences for both *C. jejuni* and *C. coli* (Supplementary Figs. 12, 13). *C. coli* causes about 10% of campylobacteriosis cases, and one clade appears to have obtained a significant portion of its genome from *C. jejuni* as it adapts to an agricultural niche[56]. We found that Cj1255, encoding a putative oxalocrotonate tautomerase, is highly conserved in *C. coli* but sporadically lost and gained in *C. jejuni* (Supplementary Figs. 12, 13). We did not detect cross-species transfer of Cj1255, but several other sORFs (CJsORF3, CJsORF39, *fedD*) appear to have been transferred between *C. jejuni* and *C. coli* (Supplementary Fig. 14). We did observe transfer signals for CJsORF27, which is present on an island (Cj1321-Cj1326 in *C. jejuni* NCTC11168) known to appear in both species[57,58]. Finally, BLASTp showed that several novel CJsORFs are in fact annotated as hypothetical proteins in other *C. jejuni* strains, including CJsORF3 (Supplementary Data File 5). Moreover, all detected CJsORF3 homologs were downstream of components of a cytochrome *bd* terminal oxidase (in NCTC11168 annotated as *cydAB*) (Supplementary Fig. 11b).

## A new small protein component of the CioAB terminal oxidase

Subcellular localization analysis using pSORTb[59] revealed that many of the novel CJsORFs have potential transmembrane helices (nine), signal peptide sequences (two), and/or are predicted to be localized to the cytoplasmic membrane (eight), suggesting they might be secreted or have functions at the cell envelope[59] (Fig. 6a and Supplementary Data File 1). This analysis again highlighted CJsORF3, annotated as a hypothetical protein outside of NCTC11168, which appears to consist mostly of a single transmembrane alpha helix. Based on its genomic context, conservation, and potential membrane localization, we focused on CJsORF3 downstream of *cydAB* (Supplementary Fig. 11b).

*E. coli* CydA and CydB form a membrane-bound complex that includes two small proteins (CydX and CydH)[60]. Literature searches also revealed that CJsORF3 was previously identified by manual inspection of regions downstream of Epsilonproteobacterial *cydB* for potential *cydX* homologs and termed *cydY*, but was not validated or studied[54]. Alignment of potential CJsORF3 homologs from diverse Epsilonproteobacteria showed strong conservation of tryptophan, proline, and tyrosine residues (Fig. 6b). Due to its similarity to the *Pseudomonas* complex, the *C. jejuni* oxidase has been re-named CioAB (cyanide insensitive oxidase)[61]. Based on these observations taken together, we renamed the CJsORF3-encoded small protein CioY and further investigated its function in *C. jejuni*.

*E. coli* CydX binds CydA (Fig. 6c, *left*)[60,62]. To determine if *C. jejuni* CioY can interact with CioA, we performed structure and complex predictions with AlphaFold2-multimer at ColabFold[63]. The results strongly supported that a CioY alpha helix interacts with CioA in a highly similar fashion as CydX-CydA (Fig. 6c, *center & right*), despite limited sequence similarity of both small proteins[54]. To validate that CioY interacts with CioA, we performed reciprocal co-immunoprecipitation (coIP) with differentially epitope-tagged small protein and potential interaction partner. Western blot analysis of eluates clearly showed that CioY binds CioA, suggesting that CioY is a new component of the CioAB terminal oxidase (Fig. 6d).

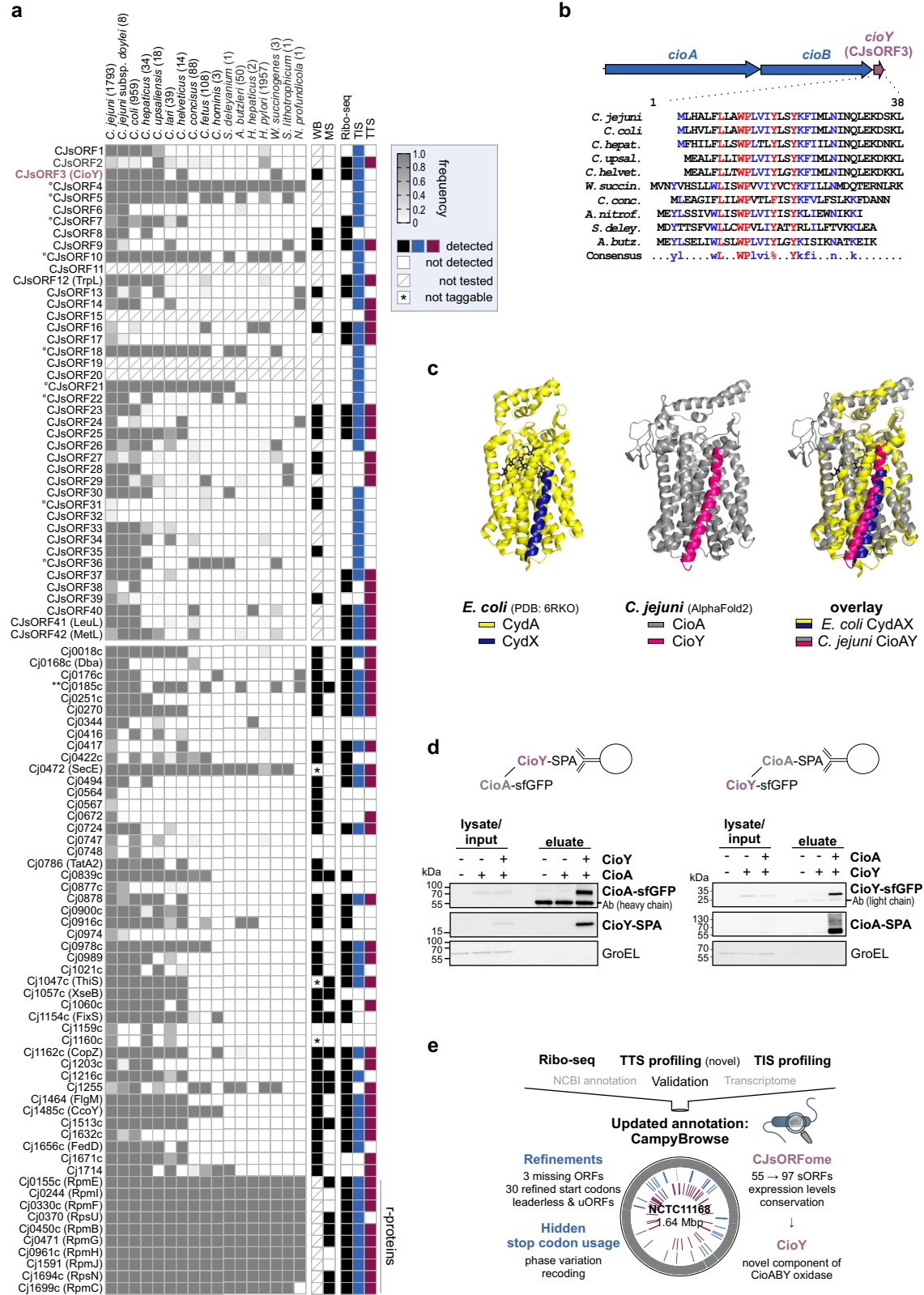

## Discussion

In this study, we integrated several Ribo-seq approaches to map the protein-coding potential of the foodborne pathogen *C. jejuni*. In addition to doubling the size of the *C. jejuni* small proteome, we also provide a comprehensive update of the *C. jejuni* ORF annotation[44] (Fig. 6e and Supplementary Fig. 15a). Our single-nt/codon resolution approach allowed us to provide evidence for previously predicted

leader peptides and leaderless ORFs, and also to re-annotate start positions of genes with roles in key *C. jejuni* phenotypes (NrfH, FliA)[28,44]. We have provided these additions to the annotation, along with translatomics datasets (Ribo-seq, TIS, and TTS profiling) and previously published transcriptome data, sRNAs, and TSS annotations[24], in our *CampyBrowse* resource (http://www.bioinf.uni-freiburg.de/~ribobase/campybrowse/overview.html) (Supplementary

**Fig. 6 | Integrated translatomics detects conserved *C. jejuni* small proteins with new functions. a** Summary of novel CJsORFs along with conservation analysis. WB: detected by western blot at any growth phase tested. Conservation is based on *C. jejuni* NCTC11168 using tBLASTn (see **Methods**). Cytopl. memb.: PSORTb prediction[59]. (°): internal in-frame. See also Supplementary Fig. 11a. **b** Genomic context and alignment of CioY sequences detected by tBLASTn in Epsilonproteobacteria. *C. jejuni*: NCTC11168. *C. coli*: RM4661. **c** Predicted complex of *C. jejuni* CioA (gray) and CioY (magenta) using AlphaFold-multimer[63] compared to *E. coli* CydX

(blue) with CydA (yellow; CydAX cryoelectron microscopy structure from[60] (PDB 6RKO)). CydBH subunits are not shown. **d** Reciprocal co-immunoprecipitation of CioY-/CioA-SPA from *C. jejuni* lysates. SPA fusions were immunoprecipitated and detected with anti-FLAG. GroEL: loading control. -/- represents untagged WT control. Ab: heavy/light chain of anti-FLAG used for coIP. Representative of *n* = 2 independent experiments. **e** Overview of *C. jejuni* translatome refinements and *CampyBrowse* resource. Source data are provided as a Source Data file.

Fig. 15b). This resource will facilitate identification of new physiological and regulatory mechanisms underlying *C. jejuni* survival and virulence.

Besides establishing TIS-profiling in *C. jejuni*, we also used Api treatment together with disome profiling to map stop codons. While our manuscript was in preparation, an *E. coli* Ribo-seq dataset, originally generated to study the mechanisms of translation inhibition by Api, was reanalyzed for sORF detection[23,29]. We also demonstrated several additional uses of TTS profiling, such as revealing stop codons (or 'non' stop codons) that are absent or not immediately apparent from the reference genome, including in phase variable ORFs, which are well-known to encode infection-related proteins in diverse pathogens. TTS profiling could reveal additional examples that are challenging to infer from the genome, such as those generated by ribosomal frameshifting. Recent MS-based proteomics analysis in *Salmonella* has led to the hypothesis that some pseudogenes, a hallmark of host-adapted strains, maintain full-length transcription and can likewise also show partial full-length translation, possibly via re-coding (via frameshifting or codon re-definition) mechanisms at or near the introduced nonsense mutation, to expand coding potential to provide opportunities in a more generalist niche[64]. TTS profiling could, in principle, be applied to identify examples of such loci, or even - under relevant conditions - measure the ratio of full-length vs. prematurely terminated translation.

Our data suggest that TTS profiling can be a generic method that can be applied to diverse prokaryotes that are sensitive to Api to map stop codons. A requirement for application of Api to Gram-negative species, including *C. jejuni*, appears to be a homolog of the SbmA transporter, as reported for Onc[22], although tightly-regulated heterologous induction of these PrAMPs as in *E. coli*[65] might circumvent this. Our study also suggests some additional guidelines for applying the approach meaningfully in other bacteria. As reported in *E. coli*, Api treatment introduced noise at *C. jejuni* TTS due to ribosome queuing (before) and readthrough (after). To circumvent this, we omitted stop codons < 25 nt downstream of annotated TTS. Also, as reported in *E. coli* and hypothesized to be due to lower affinity ribosome binding of Api even in the absence of release factors[29], Api also enriched ribosomes at some *C. jejuni* TIS. However, our parallel Onc library and disome approach (selective for ribosomes at TTS) mitigated some of the effects of this artifact. Disome footprints have been observed at stop codons in untreated eukaryotic cells[41,43], suggesting that disome profiling could be used in Api-insensitive species, although collisions at stop codons were rarely detected in untreated *E. coli*[66]. Alternatively, additional PrAMPs targeting termination could be investigated for inducing fewer Ribo-seq artifacts[67].

Comparison of annotated sORF detection by the three methods (i.e., Ribo-seq, TIS, TTS) showed that a single approach alone was not sufficient to reveal all of our independently validated benchmark sORFs, which we used to guide our genome-wide automated predictions (Fig. 6a). Based on this variable detection, we used more flexible criteria for our sORF predictions, and several of our 42 novel CJsORFs, including some validated by western blot, were predicted from only a single dataset (five TIS-only, three TTS-only). A recent re-analysis of public *E. coli* TIS/TTS datasets required sORFs to have both start and stop codon signals[23]. Our results demonstrate that while using several Ribo-seq approaches and requiring signals in all datasets can increase

confidence, more flexible criteria for sORF predictions might reveal additional, bona fide small proteins. Our approach also shows the utility of an experimentally validated sORF set to guide cutoffs to cope with high numbers of tool predictions[53], and especially that manual inspection of Ribo-seq coverage and independent validation is an essential part of Ribo-seq. We decided to manually curate the longer list of automated predictions, as our previous study with current tools showed that true sORFs can be relatively low confidence[53]. It is possible that the actual number of new sORFs is larger than our conservative set of 42; therefore, we also provide our automated predictions for future consideration.

Our Ribo-seq study, in addition to other studies, suggests that small proteomes in diverse bacteria and archaea are larger than what is currently annotated[13,17,18,21–23,68]. Taking a combined Ribo-seq approach, coupled with validation, is a strategy to generate a robust catalog of sORFs for future study. To aid in the selection of candidates for functional study, we also provide a conservation analysis of all 97 sORFs in Epsilonproteobacteria. In addition to showing that CioY is well conserved in several *Campylobacter* species, this revealed a different distribution of Cj1255 and Cj0270, two putative 4-oxalocrotonate tautomerases. Cj1255 is part of an uncharacterized tautomerase subfamily that is conserved in several important pathogenic genera (*Helicobacter*, *Yersinia*, *Neisseria*)[69]. Its substrate is currently unclear, but future examination might shed light on the metabolic strategies of pathogens carrying these small-monomer enzymes.

Current understanding of bacterial sORF function mainly places them into two categories: short proteins that interact with larger proteins/complexes to support/regulate their function, and short genes whose translation regulates adjacent ORFs. Ribo-seq revealed the small protein CioY, whose synteny suggested a function related to the CioAB terminal oxidase[54]. CioY might support oxidase maturation or activity, for example, by binding CioA and aiding assembly of/stabilizing the di-heme center as is the case for *E. coli* CydX/CydA[60,70] despite limited homology. However, heme components of *C. jejuni* CioAB are unknown[61]. Terminal oxidases appear to be rich sources of small proteins[54,60,71]. Inspection of regions lacking annotated features adjacent to operons encoding large protein complexes is, therefore, likely a fruitful strategy for revealing new functional proteins in bacteria.

While the NCBI annotation for *C. jejuni* NCTC11168 includes no ORFs below 30 aa[44], 28 of our 42 novel sORFs were ≤ 30 aa. The shortest CJsORF we confidently detected by Ribo-seq was 4 aa. However, our data suggests that many shorter sORFs ≤ 4 codons, for example, start-stop ORFs comprising only an initiating and terminating codon as reported in *E. coli*[20] might be translated (Supplementary Data file 7). The shortest functional bacterial small proteins so far include processed signaling peptides (5-10 aa)[72], antimicrobial peptides such as microcin C (7 aa)[73], and the *E. coli* CRP (cAMP receptor protein) regulating small protein SpfP (15 aa) encoded on sRNA Spot 42[72,74]. Another mode of action for many characterized sORFs is to regulate adjacent genes via their translation. In line with this, we identified the Met-codon enriched (MMYQMR) CJsORF19, a potentially new leader peptide that could regulate the expression of the downstream Cj0772c (D-methionine transport system substrate-binding

protein). Likewise, CJsORF15 (MAFY) is located upstream of a potential multidrug efflux protein (Cj0560) and could be related to translation-related control of expression of the downstream gene in the presence of translation inhibitors, as reported in, e.g., *Bacillus*[75].

At least 20 CJsORFs overlap annotated genes, e.g., CJsORF23/24, with the infection-related sRNA CJnc110[76]. However, as regulatory targets for CJnc110 are so far not clear, CJnc110 might be a small mRNA, rather than a dual (regulatory and coding) function sRNA. While in-frame, ORF-internal start codons that generate protein isoforms are an accepted phenomenon in bacterial genome architecture, the function of internal out-of-frame ORFs is enigmatic[4]. CJsORF35 (12 aa), which is encoded out-of-frame in *thiL*, appears to be more highly expressed than the parental ORF and might be encoded on a separate ORF-internal transcript with an independent function, as recently shown for several *E. coli* genes[77]. We also validated CJsORF28 (26 aa) opposite to integral membrane protein Cj1373. The function of antisense ORFs is mostly elusive[78]. More generally, it is unclear how many sORFs are nonfunctional outcomes of pervasive translation, as many are not under purifying selection or do not show biochemical features, including stability, of bona fide proteins[21,23,79]. Nonetheless, pervasive translation might provide a substrate for the evolution of new small proteins[6].

Recently, new in silico ORF prediction tools have been developed to identify small proteins in bacteria based on genome sequence alone. However, two examples (ranSEPS, smORFinder)[7,9] did not detect most of our validated CJsORFs (Supplementary Data file 1). Our validated sORF set could guide the development of the next generation of algorithms.

The rapidly expanding catalog of bacterial RNA-seq and Ribo-seq studies like ours are raising questions about how insights from these data should best be used to inform annotations. How and when should formal annotations be adjusted? How should this information be curated? While these questions are considered, we have provided our datasets and annotation refinements for inspection. Similar RNA-seq resources have been invaluable for pathogens such as *Salmonella*[80]. *CampyBrowse* is a comprehensive, unique resource with both primary transcriptome and translatome data, with extensively validated sRNAs and sORFs, and that is aimed at making re-annotations broadly available. Overall, our study has revealed new conserved and species/strain-specific features of the *C. jejuni* translatome that might contribute to house-keeping functions as well as pathogenesis and has demonstrated the utility of adding TTS profiling to identify translation features that might be not apparent from genomics alone.

## Methods

### Bacterial strains and culture conditions
All *Campylobacter jejuni* strains (Supplementary Data File 9) were routinely grown either on Müller-Hinton (MH; Becton Dickinson) agar plates or with shaking at 140 rpm in *Brucella* broth (BB; Becton Dickinson) at 37 °C in a microaerobic atmosphere (10% CO$_2$, 5% O$_2$). All *Campylobacter* media was supplemented with 10 μg/ml vancomycin. Agar was also supplemented with marker-selective antibiotics [20 μg/ml chloramphenicol (Cm), 50 μg/ml kanamycin (Kan), 20 μg/ml gentamicin (Gm), or 250 μg/ml hygromycin B (Hyg)] where appropriate. *E. coli* strains (Supplementary Data Files 9, 10) were grown aerobically at 37 °C in Luria-Bertani (LB) broth or on LB agar supplemented with the appropriate antibiotics for marker selection.

### General recombinant DNA techniques and *C. jejuni* mutant construction
All plasmids generated and/or used in this study are listed in Supplementary Data File 10. Oligonucleotide primers (Sigma) are listed in Supplementary Data File 11. DNA constructs and mutations were confirmed by Sanger sequencing (Macrogen, Microsynth). Restriction enzymes, *Taq* polymerase for validation PCR, and T4 DNA ligase were purchased from NEB. For cloning purposes, Phusion high-fidelity DNA polymerase was used (Thermo Fisher Scientific). All *C. jejuni* mutant strains (deletion, chromosomal 3 × FLAG-tagging, complementation by heterologous expression from the unrelated *rdxA* locus, Supplementary Data file 9) were constructed by double-crossover homologous recombination with DNA fragments introduced by electroporation into a *C. jejuni* strain NCTC11168 background. Details about *C. jejuni* mutant construction, as well as transformation protocols, are listed in the **Supplementary Methods**. For C-terminal epitope tagging, a 3 × FLAG, SPA as used previously[81], or superfolder GFP (sfGFP) sequence was fused to the penultimate codon of sORFs at their native locus by homologous recombination with an overlap PCR product. In some cases, SPA* was used, where the second codon of the SPA tag (ATG) was mutated to GCG.

### Total protein analysis by SDS-PAGE and western blotting
Bacterial cells were collected from cultures at the following densities (OD$_{600}$): Lag: 0.1, eExp (early exponential): 0.25, Exp (exponential): 0.5, Stat (stationary): 0.8 or from overnight cultures and resuspended in 1 × protein loading buffer (62.5 mM Tris-HCl, pH 6.8, 100 mM DTT, 10% (v/v) glycerol, 2% (w/v) SDS, 0.01% (w/v) bromophenol blue). Samples corresponding to an OD$_{600}$ of 0.1/0.2 were separated on 12% SDS-polyacrylamide (PAA) gels, or on 16% separating/4% stacking Tricine-SDS-PAGE gels without urea[82] (stacking – 30 V, separation 50-120 V). Separated proteins were transferred to a nitrocellulose membrane by semidry blotting (Peqlab). After transfer, membranes were stained for 5 min in Ponceau S (Serva) to visualize transferred proteins (0.25% w/v Ponceau S, 5% acetic acid). Membranes were then blocked with 10% (w/v) milk powder in TBS-T (Tris-buffered saline-Tween-20) and incubated overnight with the primary antibody in 3% BSA/TBS-T (monoclonal mouse anti-FLAG, 1:1,000; Sigma-Aldrich, #F1804-1MG, RRID:AB_262044 or monoclonal mouse anti-GFP, 1:1,000, Roche #11814460001, RRID:AB_390913) at 4 °C. Washed membranes (TBS-T) were then incubated with secondary antibody (sheep polyclonal, anti-mouse IgG horseradish peroxidase (HRP) conjugate, 1:10,000 in 3% BSA/TBS-T; GE Healthcare, #RPN4201). Blots were developed using enhanced chemiluminescence reagent on an ImageQuant LAS-4000 Imager (GE Healthcare, version 1.3, build 1.3.0.134). An antibody specific for GroEL (rabbit polyclonal, 1:10,000 in 3% BSA/TBS-T; Sigma-Aldrich, #G6532-5ML, RRID:AB_259939) with an anti-rabbit IgG (goat polyclonal, 1:10,000 in 3% BSA/TBS-T; GE Healthcare, #RPN4301, RRID:AB_2650489) secondary antibody was used as a loading control. For size estimation, a Spectra™ Multicolor Low Range Protein Ladder (Thermo Fisher Scientific) or Prestained Protein Marker (Thermo Fisher Scientific) were loaded on Tricine-PAGE or SDS-PAGE gels, respectively. Western blot analysis of at least two independent biological replicates was performed, and a protein was called as translated when it was detected in both replicates.

### Growth and cell harvest for ribosome profiling
For Ribo-seq, *C. jejuni* NCTC11168 WT or Δ*cmeB* mutant strains were grown in BB to log phase (OD$_{600}$ ~ 0.4-0.5). Full details are provided in the **Supplementary Methods**. For Ribo-seq with Cm, *C. jejuni* NCTC11168 WT cells were treated with 1 mg/ml Cm for 5 min at 37 °C under microaerobic conditions, followed by immediate chilling for 10 min by mixture with an equal volume of crushed ice containing 1 mg/ml Cm. Cells were recovered by centrifugation and snap-frozen in liquid N$_2$. For the TIS(Ret) experiment, Δ*cmeB* mutant cells in log phase were treated with 12.5 μg/ml Ret (Sigma CDS023386) for 10 min, followed by recovery by fast-filtration using a 0.45 μm polyethersulfone membrane and snap freezing in liquid N$_2$ as done previously[17,53]. For the TIS/TTS experiment, *C. jejuni* NCTC11168 WT cultures were treated with peptide (NovoPro BioSciences, Shanghai; 50 μM final concentration, 10 min), immediately chilled in an ice bath with swirling for 3 min, recovered by centrifugation, and snap-frozen in liquid N$_2$. MIC determination for Ret, Onc, and Api is described in the **Supplementary**

**Methods**. Three independent biological replicates (cultures) were used for translatomics experiments.

## Processing of cell pellets and isolation of monosomes and footprints

Processing of cell pellets was performed generally as described previously[17,53] as follows. Cm was omitted from lysis buffers for TIS and TTS experiments but used at 1 mM for Ribo-seq(Cm). PNP-GMP (guanosine 5′-[β,γ-imido]triphosphate, 3 mM, Sigma) was included in the lysis buffer of TIS(Ret) samples. Frozen cells were mixed with frozen lysis buffer (100 mM NH$_4$Cl, 25 mM MgCl$_2$, 20 mM Tris-HCl, pH 8, 0.1% NP-40, 0.4% Triton X-100) supplemented with 50 U DNase I (Thermo Fisher Scientific) and 500 U RNase inhibitor (moloX, Berlin). For Ribo-seq(Cm) and TIS(Ret), cells were lysed in an MM-400 metal ball mill (Retsch) for 5 rounds at 15 Hz for 3 min with chilling of the mill in liquid N$_2$ between rounds. Lysates were thawed by incubation in a water bath at 30 °C for 2 min and centrifuged at $10,000 \times g$ for 5 minutes. For the TIS(Onc)/TTS experiment, cells were lysed using a FastPrep system (MP Biomedical) with lysing Matrix B, speed 4, for $3 \times 20$ s. Clarified lysates (approximately 15 A$_{260}$ units) were digested with 800 U/A$_{260}$ MNase (New England Biolabs) for 1.5 h (25 °C, shaking at 1450 rpm). Digests were stopped with EGTA (final concentration, 6 mM), immediately loaded onto 10−55% (w/v) sucrose density gradients freshly prepared in sucrose buffer (100 mM NH$_4$Cl, 25 mM MgCl$_2$, 5 mM CaCl$_2$, 20 mM Tris-HCl, pH 8, 2 mM dithiothreitol), and centrifuged (35,000 rpm, 2.5 h, 4 °C) in a Beckman Colter Optima L-80 XP ultracentrifuge and SW40 Ti rotor. Gradients were fractionated (Gradient Station ip, Biocomp), and the 70S monosome fraction (identified by following fraction A$_{260}$) was immediately frozen in liquid N$_2$.

## RNA isolation and cDNA library preparation for translatomics

RNA was extracted from fractions or cell pellets for total RNA using hot phenol-chloroform-isoamyl alcohol or hot phenol, respectively, as performed previously[83] and described in the **Supplementary Methods**. Total RNA was digested with DNase I (ThermoFisher), depleted of rRNA (Ribo-zero Bacteria, Illumina), and fragmented (Ambion 10 × RNA Fragmentation Reagent) according to the manufacturer's instructions. Monosome RNA and fragmented total RNA was size-selected (26-34 nt) on 15% polyacrylamide/7 M urea gels using the RNA markers NI-19 & NI-20[36] as a guide and extracted from gel slices via homogenization, incubation in 300 mM sodium acetate, pH 5.5, 1 mM EDTA, 0.25% wt/vol SDS, and precipitation with 1/10 volume sodium acetate, pH 5.2 and 1 volume of isopropanol with 0.5 µl Glycoblue. Libraries were prepared by vertis Biotechnologie AG (Freising, Germany) using a small RNA protocol (Ribo-seq(Cm) or TIS(Ret) experiment) or an Adapter Ligation protocol (TIS(Onc)/TTS experiment) and sequenced on a NextSeq500 instrument (high-output, 75 cycles) at the Core Unit SysMed at the University of Würzburg.

## General processing of ribosome profiling data

Sequencing data was processed and analyzed with HRIBO (version 1.4.4)[52]. The reference genome sequence NC_002163.1 (ASM908v1)[44,47] was used in combination with a custom annotation file including previously annotated 5′UTRs and sRNAs[24] for Ribo-seq data analysis. To generate this file, the NCBI annotation (2014-03-20) was combined with a 5′UTR and sRNA annotation generated based on dRNA-seq data[24]. In brief, primary transcriptional start sites in the NCTC11168 genome, as well as the strand information (Supplementary Table S4) was used to calculate 5′UTR end positions based on UTR lengths to identify 5′UTR regions. The resulting 5′UTR regions were exported in GFF3 format. To generate the sRNA annotation file, the annotation table for strain NCTC11168 from[24] (Table S11) was used. CJas_Cj0168c, not detected by NB, was excluded from the final sRNA annotation file. The three .gff files were combined and attribute columns were updated to fit the GFF3 standard. Furthermore, four housekeeping RNAs

(SRP, RnpB, tmRNA, and TPP riboswitch) were added to the custom annotation (also from Supplementary Table S11[24]). This annotation (generated 2018-12-03) can be downloaded from GEO (GSE208756) or *CampyBrowse*. A comparison of the NCBI genome annotation used for analysis (2014-03-20) to the one currently available at NCBI (updated 2021-09-11, downloaded 2022-11-07) revealed that they are identical except for two gene fusions that were replaced by single pseudogenes (*uxaA'* (Cj0482)/*uxaA* (Cj0483), now Cj0483 pseudogene and *metC'* (Cj1392)/*metC* (Cj1393), now Cj1392 pseudogene). Summary statistics for sequencing libraries can be found in Supplementary Data File 12.

## Ribo-seq-based ORF prediction

ORFs were called from Ribo-seq libraries using two prediction tools included in HRIBO (version 1.4.4): DeepRibo[51] and Reparation_blast (version 1.0.9), an adapted version of REPARATION[50], which uses blast instead of usearch (https://github.com/RickGelhausen/REPARATION_blast)[53]. Predictions were generated from normal Ribo-seq data from all datasets ((Ribo-seq (Cm), control (no drug) libraries from the TIS/TTS experiments). Three replicates were used for Ribo-seq(Cm) and TIS(Ret) experiment data, while only the first available replicate for the TIS(Onc)/TTS experiment was used to generate predictions. The results of both DeepRibo and REPARATION from all analyzed experiments/replicates were aggregated into Excel tables with additional information for each detected ORF, e.g., expression, translational efficiency, DeepRibo score for subsequent filtering (see section on Filtering below).

## Detection of TIS/TTS sites and associated ORFs

For the detection of TIS/TTS, we adapted previous peak detection methods[13,22] as follows. Briefly, we used single-nucleotide mapping coverage files generated by HRIBO (version 1.4.4)[52] for read 3' ends and read lengths providing the sharpest enrichment near the expected offset positions with respect to the start/stop codon within each experiment, based on metagene analysis[30]. Selected mapping files and offsets were then used for peak detection at start codons (ATG/GTG/TTG) at intervals of 5 nt. For each interval with a non-zero peak height, the next in-frame stop codon was identified to generate a corresponding ORF (of any length). For TTS-based sORF detection, a similar approach was taken (TGA/TAA/TAG codons), except potential sORFs with a stop codon within 25 nt of an annotated stop codon were omitted to exclude predictions arising from read-through and stop codons with longest potential length ≤ 71 codons or shortest potential length ≥ 11 codons were investigated manually for start codon peaks, due to ambiguity in assigning start codons. For details, including a list of read-lengths and offsets for all experiments, see the **Supplementary Methods**. All programming scripts are available at https://github.com/RickGelhausen/StartStopFinder (version 1.0.0).

## Filtering of Ribo-seq/TIS/TTS ORF predictions

To generate a short list of candidate sORFs (11-71 codons) from HRIBO Ribo-seq-based predictions, we applied the following expression cut-offs (mean TE and RNA-seq RPKM (within an experiment) ≥1 and ≥30, respectively, see also Supplementary Fig. 7a, d). Candidates were required to be detected by REPARATION or by DeepRibo (with a Score > 0) in at least one replicate of the experiment. We also required candidates to be detected in a Ribo-seq library for at least two out of the three experiments (Ribo-seq(Cm), TIS(Ret), and TIS(Onc)/TTS). To generate a list of sORF candidates based on TIS for curation, we applied the following cutoffs (see also Supplementary Fig. 7b, d): ORF length between 11-71 codons, peak height (Ret or Onc TIS library) ≥ 20, a peak in the TIS library only or a log2FC (fold-change) ≥1 in at least two replicates (TIS(Ret) experiment) or the single TIS/TTS(Onc/Api) replicate used for predictions. For TIS predictions, we also manually inspected the top 25 candidates (detected in all three replicates with a TIS peak only or log2FC > 1, sorted by TIS peak height; resulting in total

in 40 candidates) shorter than the DeepRibo/REPARATION length cutoff of 11 codons (2-10 codons). To filter TTS sites, we applied the following criteria (see also Supplementary Fig. 7d). Because of uncertainty in assigning start codons, predictions were associated with a longest and a shortest potential ORF. Candidate sORFs were filtered for those with the longest potential ORF ≤ 71 codons and the shortest potential ORF ≥ 11 codons. The minimum TTS peak height was required to be ≥ 5 reads. For TTS(Mono), all candidates with a peak at the same position in the corresponding TIS(Onc) library and all internal out-of-frame candidates were excluded.

### Manual curation of Ribo-seq/TIS/TTS-based sORF predictions

After the above-described filtering of sORF predictions, those remaining were manually curated essentially as previously[14,53,84], details of which are described in the **Supplementary Methods**. For manual curation, three replicates of TIS(Onc) and TTS were included. Reproducibility between those replicates is compared in Supplementary Data File 13.

### Additional bioinformatics and analysis

RBS motifs were predicted using MEME (version 5.4.1, RRID:SCR_001783)[85]. Subcellular localization of small proteins was predicted using pSORTb v3.0 (RRID:SCR_007038) with default parameters for Gram-negative bacteria[59]. Our interactive web-based genome-browser to visualize coverage files and re-annotations used in this study was established using JBrowse (v2.11.1) (RRID:SCR_001004)[86].

For sORF conservation analysis in Epsilonproteobacteria, we downloaded publicly available assembled genome sequences for 17 species from the National Center for Biotechnology Information GenBank or RefSeq databases (RRID:SCR_003496) (RRID:SCR_002760) (before Jan 2023). Strains and genome accessions used for conservation analysis are listed in Supplementary Data File 8. After genome quality estimation using checkM (v1.1.3)[87] (RRID:SCR_016646) and discarding genomes where several small ribosomal proteins were absent, we obtained a total of 5209 high-quality genomes (completeness > 95% and contamination < 5%) for further analysis.

We also performed MLST (multilocus sequence typing) and phylogenetic analysis for two species, *C. coli* and *C. jejuni*. MLST sequence types and clonal complexes were obtained by scanning the genome sequences against the PubMLST database using mlst (v2.23.0) (https://github.com/tseemann/mlst)[88] (RRID:SCR_010245). Maximum-likelihood phylogenetic trees were constructed using FastTree (v2.1.10)[89] (RRID:SCR_015501) based on core genome (defined as regions present in > 99% isolates) single nucleotide polymorphisms (SNPs) as previously described[90,91]. Briefly, the assemblies were aligned against the reference genomes (*C. coli*: NC_022347.1, CVM N29710; *C. jejuni*: NC_002163.1, NCTC11168) using MUMmer (v3.23)[92] (RRID:SCR_018171) to generate whole genome alignments. SNP calling was performed using *SNP-sites* (v2.5.1)[93] (RRID:SCR_022265) based on the alignment. Repetitive regions of the reference genome were identified using TRF (v4.07b)[94] (RRID:SCR_022193) and self-aligning by BLASTn (v2.11.0 +)[95] (RRID:SCR_001598). SNPs located in repetitive regions were excluded from phylogenetic analysis.

For sORF conservation analysis, we used tBLASTn (v2.11.0 + )[95] (RRID:SCR_011822) to detect the presence or absence of sORFs using an E-value (Expect value) cutoff of ≤ 1 and word size of 3, with low complexity filtering turned off (-seg no). All other parameters were set to default. An sORF was considered present if both the coverage and identity were ≥ 50%. Those that could not be re-detected in the reference strain NCTC11168 were excluded from further analysis. The degree of conservation was reported as the frequency in the species/strain. To inspect potential cross-species (*C. jejuni* & *C. coli*) transfer of sORFs, sequences were aligned with MAFFT version 7 (v7.505)[96] (RRID:SCR_011811) and used to generate a phylogenetic tree with FastTree[89] (RRID:SCR_015501).

Annotated homologs were detected by BLASTp at NCBI (RRID:SCR_001010) using default parameters. For a "yes" match, both an identical length and amino acid sequence was required. For a partial match, an E-value < 100 was still required, but the length requirement was omitted (*i.e.*, the matching ORF could be the longer parental/overlapping ORF for an internal in-frame candidate).

### Mass spectrometry based proteomics

Small proteins were identified in *C. jejuni* soluble protein extracts, generated by lysis using a FastPrep Homogenizer (MP-Biomedicals), by MS from bacteria growing in the log phase in BB. Two different techniques for pre-fractionation of proteins were applied: (i) separation of soluble proteins by one dimensional (1D) SDS-PAGE and in-gel digestion with trypsin or chymotrypsin (see "gel-based approach" described in ref. 18, or (ii) fractionation of proteins on a GELFREE 8100 fractionator (Expedeon) with 10% Tris-acetate cartridges and trypsin digestion. Digestion of proteins using a GELFREE 8100 fractionator was performed in protein low binding tubes (Eppendorf, Hamburg, Germany) using the Single-Pot Solid-Phase-enhanced Sample Preparation technique described previously[97] with modifications as described in the **Supplementary Methods**. Peptide fractions were analyzed using the Orbitrap Fusion MS coupled to a Dionex Ultimate 3000 nHPLC system (Thermo Fisher Scientific Inc., Waltham, Massachusetts, USA) as described previously[18] with modifications. The full procedure is described in the **Supplementary Methods**.

For identification of small proteins based on MS/MS data, we used the fully automated bacterial proteogenomics workflow SALT & Pepper (https://gitlab.com/s.fuchs/pepper)[18], which includes protein database generation, database searching, peptide-to-genome mapping, and result interpretation. MS- and MS/MS-data of all samples were searched by MaxQuant (Max Planck Institute of Biochemistry, Martinsried, Germany, www.maxquant.org, version 1.5.2.8, RRID:SCR_014485) against a database with *C. jejuni* annotated protein sequences from NCBI (downloaded on 01-09-2020) and sORFs that were predicted using our Ribo-seq data and a translational database (TRDB) of the full coding potential of the *C. jejuni* genome generated by six-frame translation from stop codon to stop codon with a minimum length of 9 aa generated by SALT (https://gitlab.com/s.fuchs/pepper). Three independent biological replicates were used for analysis. Full details of MS-based proteomics can be found in the **Supplementary Methods**.

### Microscopy

For FITC (fluorescein isothiocyanate) labeling of *C. jejuni*, ~ 1.5 - 2 × 10^8 cells, grown in log phase in BB + 10 μg/ml vancomycin, were harvested by centrifugation (6600 × g, 5 min at room temperature) and washed once with 1 × PBS. Freshly prepared 10 mg/ml FITC (Sigma) in 100% ethanol was diluted to 0.1 mg/ml in 1 × PBS. Bacteria were resuspended in this solution for 30 min (37 °C, microaerophilic conditions, shaking), washed twice with 1 × PBS, and fixed with 4% paraformaldehyde for 1 hour at room temperature in the dark. After washing once with 1× PBS, cells were resuspended in 1× PBS and placed on an agarose pad (1% agarose in 1 × PBS) and imaged with a laser scanning Leica TCS SP5 II confocal microscope (Leica Microsystems).

### Motility assays

Liquid cultures of each strain were grown to log phase in BB media + 10 μg/ml vancomycin. Next, 1 μl of bacterial culture was inoculated into a soft-agar plate (BB broth + 0.4% Difco agar, 10 μg/ml vancomycin). Plates were incubated right-side-up at 37 °C under microaerobic conditions until halo formation could be observed (approximately 24 hrs post-inoculation). Each halo radius of technical triplicates was measured two times and averaged to give the mean swimming distance per strain. Motility assays were performed in three independent biological replicates. Student's *t*-test (unpaired, two-tailed, using GraphPad Prism 7) was used to assess significance.

**Co-immunoprecipitation (coIP) for investigation of protein-protein interactions**

Lysates were prepared from *C. jejuni* strains, grown to log phase, carrying chromosomally epitope-tagged versions of CioA and/or CioY (CioA-SPA & CioY-sfGFP and reciprocal version CioY-SPA & CioA-sfGFP), with a FastPrep system (MP Biomedical, matrix B and lysis buffer with 1% DDM (n-dodecyl-B-D-maltoside). Lysates of the untagged wild-type strain as well as the corresponding sfGFP-only tagged strains (CioA-sfGFP or CioY-sfGFP alone) were used as a control for unspecific binding. SPA-tagged protein was pulled down from clarified lysates with an anti-FLAG antibody (Sigma-Aldrich, #F1804-1MG, RRID:AB_262044) bound to Protein A-Sepharose beads (Sigma-Aldrich, #P6649). Co-purification was investigated by western blot with an anti-GFP antibody (Roche #11814460001, RRID:AB_390913). Two independent biological replicates were performed. Full details can be found in the **Supplementary Methods**.

**Protein complex and structure prediction with AlphaFold2**

Structural predictions of protein complexes were performed at ColabFold (https://colab.research.google.com/github/sokrypton/ColabFold/blob/main/AlphaFold2.ipynb#scrollTo=KK7X9T44pWb7, accessed 2022-03-26) using AlphaFold2 and AlphaFold2-multimer[63]. Standard settings were used (msa_mode: MMseqs2 (UniRef+Environmental); pair_mode: unpaired+paired; model_type: auto; num_recycles: 3). The best-ranked structure prediction was selected, where pLDDT values for all proteins in the complex were evaluated in addition. Graphics of structural predictions, structures from PDB (6RKO), as well as overlays were generated in Pymol (version 2.5.2, RRID:SCR_000305).

**Reporting summary**

Further information on research design is available in the Nature Portfolio Reporting Summary linked to this article.

## Data availability

Ribo-seq data generated in this study have been deposited at the NCBI Gene Expression Omnibus (GEO) under the accession GSE208756. The mass spectrometry proteomics data have been deposited to the ProteomeXchange Consortium (http://proteomecentral.proteomexchange.org) via the PRIDE partner repository (https://www.ebi.ac.uk/pride/archive)[98] with the identifier PXD036790. Differential RNA-seq data[24] used in this study are available at GEO under the accession GSE38883. The cryo-EM structure of the *E. coli* cytochrome *bd*-I oxidase is available at PDB under the accession 6RKO. *CampyBrowse* resource is available at http://www.bioinf.uni-freiburg.de/~ribobase/campybrowse/overview.html and http://www.bioinf.uni-freiburg.de/ribobase. Source data are provided in this paper.

## Code availability

The HRIBO pipeline used in this study (version 1.4.4) is available at https://github.com/RickGelhausen/HRIBO. The adapted version of REPARATION (REPARATION_blast, version 1.0.9) is available at https://github.com/RickGelhausen/REPARATION_blast. All programming scripts for TIS and TTS predictions (StartStopFinder, version 1.0.0) are available at https://github.com/RickGelhausen/StartStopFinder. For the identification of small proteins based on MS/MS data, we used SALT & Pepper available at https://gitlab.com/s.fuchs/pepper.

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

## Acknowledgements

We thank Gaurav Dugar, Sandy Pernitzsch, and Lydia Hadjeras for discussions on Ribo-seq protocols, as well as Thorsten Bischler and Diego Gelsinger for guidance on Ribo-seq and TIS data analysis. We thank Daniel Wilson for discussions regarding TTS profiling and Julian Langner for fruitful discussions regarding oxidase complexes as well as for providing reagents. We are also grateful to Susan Gottesman for insightful comments on the manuscript. Small protein and Ribo-seq research in the Sharma laboratory is supported by an individual project grant within the Deutsche Forschungsgemeinschaft (DFG) priority program SPP2002 "Small proteins in prokaryotes, an unexplored world" to C.M.S. (SH580/8-1 and SH580/8-2). This work was also supported by the Z2 Central Project "Ribosome Profiling and Bioinformatics" within the SPP2002 (awarded to C.M.S. SH580/7-1 and 7-2 and R.B. BA2168/21-2). Moreover, R.B. was supported by funds from Germany's Excellence Strategy (CIBSS - EXC-2189 - Project ID 390939984). K.F. was supported by a grant of the German Excellence Initiative to the Graduate School of Life Sciences, University of Würzburg. Computational resources were provided by the BMBF-funded de.NBI Cloud within the German Network for Bioinformatics Infrastructure (de.NBI) (031A532B, 031A533A, 031A533B, 031A534A, 031A535A, 031A537A, 031A537B, 031A537C, 031A537D, 031A538A) to R.B. MS-based small protein analyses were supported by an individual project grant within the DFG priority program SPP2002 "Small proteins in prokaryotes, an unexplored world" to S.E. and S.F. (EN 712/4-1), by a grant of the GRK PROCOMPAS (DFG) to S.E., and an institutional grant (INST 188/365-1 FUGG DFG) to S.E. C.Y. is supported by the Youth Innovation Promotion Association, Chinese Academy of Sciences (No. 2022278) and S.L.S. by RFIS-II funding from the National Natural Science Foundation of China (No. 32250610209).

## Author contributions

K.F., S.L.S. and C.M.S. designed the research. K.F., S.L.S., E.F., P.K. and A.K. performed lab work. K.F., S.L.S., R.G., A.K., M.K., S.F., C.Y. and F.E. analyzed data. K.F., S.L.S. and C.M.S. interpreted the data and wrote the manuscript, which all co-authors revised. R.B., S.E., D.F. and C.M.S. supervised research and provided resources.

## Funding

## Competing interests
The authors declare no competing interests.
