## [Peer Review file · Nature Communications]

Complementary Ribo-seq approaches map the translome and provide a small protein census in the foodborne pathogen *Campylobacter jejuni*

Corresponding Author: Professor Cynthia Sharma

Version 0:

Reviewer comments:

Reviewer #1

(Remarks to the Author)

I appreciate the significant effort the authors have made to address the reviewers' comments both in the point-by-point response and in the changes to the manuscript. I think the streamlined text, the more detailed information about the curation, the conservation analysis and other new figures have significantly improved the paper. The data presented will be very valuable to the *Campylobacter* community.

Reviewer #2

(Remarks to the Author)

The authors have been very responsive to the reviewers' comments. I appreciate their efforts to give a more detailed description of the manual curation, and to provide the full list of sORFs from the automated pipelines. I would encourage the authors to automate their curation process for future work, but for this study I think the information provided is sufficient.

Reply to reviewers:

Reviewer #1 (Remarks to the Author):

I appreciate the significant effort the authors have made to address the reviewers' comments both in the point-by-point response and in the changes to the manuscript. I think the streamlined text, the more detailed information about the curation, the conservation analysis and other new figures have significantly improved the paper. The data presented will be very valuable to the Campylobacter community.

We thank Reviewer 1 for the positive response and acknowledgements of the revised manuscript.

Reviewer #2 (Remarks to the Author):

The authors have been very responsive to the reviewers' comments. I appreciate their efforts to give a more detailed description of the manual curation, and to provide the full list of sORFs from the automated pipelines. I would encourage the authors to automate their curation process for future work, but for this study I think the information provided is sufficient.

We thank Reviewer 2 for the positive feedback on our revised manuscript.